# One Loss for All:
# Deep Hashing with a Single
# Cosine Similarity based Learning Objective

**Jiun Tian Hoe**[1*]     **Kam Woh Ng**[2,3*]     **Tianyu Zhang**[4]
**Chee Seng Chan**[1†]     **Yi-Zhe Song**[2,3]     **Tao Xiang**[2,3]

[1]CISiP, Universiti Malaya, Malaysia
[2]CVSSP, University of Surrey, U.K.
[3]iFlyTek-Surrey Joint Research Centre on Artificial Intelligence
[4]Geek+, China

## Abstract

A deep hashing model typically has two main learning objectives: to make the learned binary hash codes discriminative and to minimize a quantization error. With further constraints such as bit balance and code orthogonality, it is not uncommon for existing models to employ a large number (>4) of losses. This leads to difficulties in model training and subsequently impedes their effectiveness. In this work, we propose a novel deep hashing model with only *a single learning objective*. Specifically, we show that maximizing the cosine similarity between the continuous codes and their corresponding *binary orthogonal codes* can ensure both hash code discriminativeness and quantization error minimization. Further, with this learning objective, code balancing can be achieved by simply using a Batch Normalization (BN) layer and multi-label classification is also straightforward with label smoothing. The result is an one-loss deep hashing model that removes all the hassles of tuning the weights of various losses. Importantly, extensive experiments show that our model is highly effective, outperforming the state-of-the-art multi-loss hashing models on three large-scale instance retrieval benchmarks, often by significant margins. Code is available at `https://github.com/kamwoh/orthohash`.

## 1 Introduction

A key building block of a real-world large-scale image retrieval system is hashing. The objective of image hashing is to represent the content of an image using a binary code for efficient storage and accurate retrieval. Recently, deep hashing methods [48, 23] have shown great improvements over conventional hashing methods [46, 15, 16, 22, 36, 37, 19]. Furthermore, deep hashing methods can be grouped by how the similarity of the learned hashing codes are measured, namely pointwise [49, 54, 40, 12, 50], pairwise [25, 23, 5, 4], triplet-wise [45, 32], or listwise [52]. Among them, pointwise methods have a $O(N)$ computational complexity, whilst the complexity of the others are of at least $O(N^2)$ for $N$ data points. This means that for large-scale problems, only the pointwise methods are tractable [49]. They are thus the focus of most recent studies.

A deep hashing neural network naturally has multiple learning objectives. Specifically, given an image input, the network outputs a continuous code (feature vector) which is then converted into a binary

---

[*]equal contribution.

[†]corresponding author (*cs.chan@um.edu.my*).

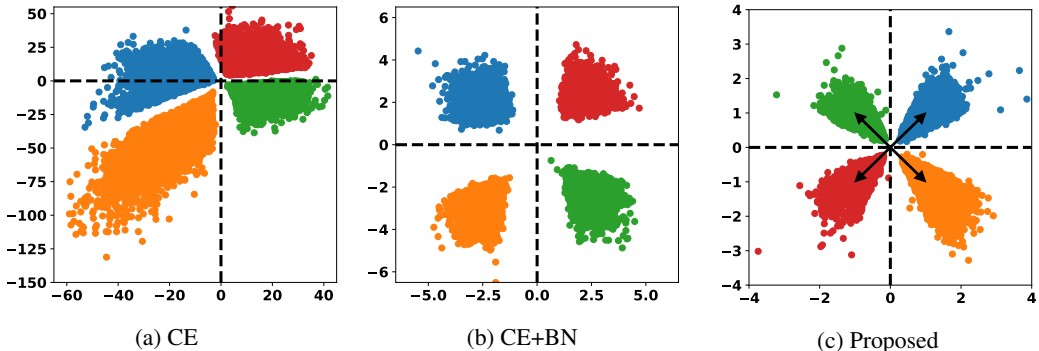

| (a) CE | (b) CE+BN | (c) Proposed |

Figure 1: We train a simple CNN model on CIFAR10 with only first 4 classes and 2-bits. The continuous codes $\mathbf{v}$ are visualized before *sgn*. **(Left)** The model is trained with cross entropy **(CE)** only. Although it can separate the 4 classes in Euclidean space, the output is not bounded and thus indicating high quantization error and sub-optimal in the Hamming space. **(Middle)** By appending a batch normalization **(BN)** layer after $\mathbf{v}$, the hash codes are now balanced. **(Right)** Now the model (proposed) is trained to maximize the cosine similarity between $\mathbf{v}$ and its corresponding binary target $\mathbf{o}$. The black arrows are the **binary orthogonal target**, denoted as $\mathbf{o}$ for each class. It can be seen that the continuous codes exhibit lower intra-class variance and quantization error as compared with the **CE+BN** models (middle).

hash code using a quantization layer (usually a *sign* function). There are thus two main objectives. First, the final model output, i.e., the binary codes must be discriminative, meaning the intra-class hamming distances are small, while the inter-class ones are big. Second, a quantization error minimization objective is needed to regularize the continuous codes. But the learning is constrained by the vanishing gradient problem caused by the quantization layer. Although the problem can be avoided by deploying some relaxation schemes [5, 23, 25], these schemes often produce sub-optimal hash codes due to the introduction of quantization error (see Figure 1). Hence, most recently deep hashing methods [41, 25, 4, 53, 50] has an explicit quantization error minimization learning objective.

Having these two main objectives/losses are still not enough. In particular, to ensure the quality of hash codes, many other losses are employed by existing methods. These include bit balance loss [53, 49, 40], weights constraints to maximize Hamming distance [54], code orthogonality [31, 32]. Further, losses are designed to address the vanishing gradient problem caused by the sign function used to obtain binary codes from the continuous ones [41, 40, 27]. As a result, the state-of-the-art hashing models typically have a large number (>4) losses. This means difficulties in optimization which in turn hamper their effectiveness.

In this work, for the first time, a deep hashing model with a single loss is developed which removes any needs for loss weight tuning and is thus much easier to optimize. As mentioned earlier, a deep hashing model needs to be trained with at least two objectives, namely binary code discriminativenss and quantization error minimization. So how could one use one loss only? The answer lies in the fact that the two objectives are closely related and can be unified into one. More concretely, we show that both objectives can be satisfied by maximizing the cosine similarity between the continuous codes and their corresponding *binary orthogonal target*, which can be formulated as a cross-entropy (CE) loss. Our model, dubbed **OrthoHash** has one loss only which maximizes the cosine similarity between the $L_2$-normalized continuous codes and *binary orthogonal target* to maximize inter-class Hamming distance and minimize quantization error simultaneously. We show that this single unifying loss has a number of additional benefits. First, we can leverage the benefit of margin [42, 10] to further improve the intra-class variance. Second, since conventional CE loss only works for single-label classification, we can easily leverage Label Smoothing [38] to modify the CE loss to tackle multi-labels classification. Finally, we show that code balancing can now be enforced by introducing a batch normalization [17] (BN) layer rather than requiring a different loss. Extensive experiment results suggest that on conventional category-level retrieval tasks using ImageNet100, NUS-WIDE and MS-COCO, our model is on par with the SOTA. More importantly, on the large-scale instance-level retrieval tasks, our method achieves the new SOTA, beating the best results obtained so far on GLDv2, $\mathcal{R}$Oxf and $\mathcal{R}$Paris by 0.6%, 9.1% and 17.1% respectively.

## 2 Related Work

**Hashing methods.** Conventional hashing methods can be categorized into many streams. Data-independent methods such as Locality-sensitive Hashing (LsH) [16, 13], and its kernelized version (KLsH) [22] have contributed many of the fundamental concepts for hashing such as the requirement of *code balance*, *uncorrelated bit*, and *similarity preserving*. In contrast, data-dependent methods [46, 21, 15, 19, 36, 37] aim to learn hash codes that are more compact yet more dataset-specific [7]. Recently, deep learning based hashing methods [25, 48, 23] dominated the hashing research due to the superior learning ability of DNN. Various learning objectives are developed to learn hash codes using a training dataset. The objective functions include i) task learning objective which can be further categorized into pointwise [49, 54, 40, 41, 12, 50], pairwise [5, 25, 23], triplet-wise [45, 32], listwise [52] and unsupervised [27, 14]; ii) quantization error minimization such as the loss designed to minimize the $p$-norm (usually $p = 2$) between continuous codes and hash codes; iii) code balancing [27, 40]. We refer readers to learning to hashing surveys [44, 43, 11] for more detailed review.

**Binary optimization.** Hashing is a NP-hard binary optimization problem [46], and is prone to the vanishing gradient problem due to the discrete and non-differentiable binary hash functions. Early methods solved the problem by discarding the discrete constraints (e.g., designing a penalty loss term to generate feature as binary as possible [25, 23]; solve with continuous relaxation, i.e., to optimize in a continuous space using *sigmoid* or *tanh* for approximation [5]). Some methods also utilized coordinate descent method in the training [28, 24]. Nevertheless, these methods have increased the complexity of learning due to need for tuning of hyper-parameters balancing different learning objectives.

**Bypassing vanishing gradient.** Greedy Hash [41] designed a new coding layer which uses the sign function in the forward pass to generate binary codes, and gradients are backpropagated using straight-through estimator [1] during optimization. [27] designed a parameter-free coding layer – Bi-half, to maximize the bit capacity by shifting the network output by *median* (each bit can have a 50% chance of being $+1$ or $-1$) . These methods typically requires the modification of computational graphs, in the sense that the original graph is no longer end-to-end trained, hence further complicates the original optimization objective. Ours on the other hand incorporates a neat one-loss design that removes all such complications.

**Learning hash codes with pre-defined target.** Deep Polarized Network (DPN) [12] used a random assignment scheme to generate target vectors with maximal inter-class distance, then optimized with hinge-like polarized loss. Central Similarity Quantization (CSQ) [50] uses Hadamard matrix as "hash centers", then optimized with binary cross entropy. Both methods have similar overall objective, i.e., the continuous codes are learned to be as similar as the target vectors (or "hash centers"). Our model also employs a hash target, but uniquely it is used in a single cosine similarity based single objective.

**Cosine similarity.** According to [6], which is a theoretical analysis for Locality-sensitive Hashing (LsH) [16, 13], if two samples have high angular similarity, then we have high probability of obtaining the same hash codes as well. Hence, while most works focus on hashing images with various constraints, we reformulate the problem of deep hashing in the lens of cosine similarity. By following the same principle, a similar work is done by [2] which described the hashing problem under pairwise constraint while our work describes the problem under pointwise constraint. As inspired by [51, 14] which utilize cosine similarity to find closest approximate binary or ternary representation, we also interpret the quantization error in terms of cosine similarity. Moreover, deep hypersphere embedding learning methods (e.g., SphereFace [35], CosFace [42] and ArcFace [10]) imposed discriminative constraints on a hypersphere manifold and proposed to improve decision boundary by cosine or angular margin. Inspired by them, we also leverage the benefit of margin to improve intra-class variance.

## 3 OrthoHash: One Loss for All

In Section 3.1, we reformulate the problem of deep hashing in the lens of cosine similarity, i.e., interpreting both Hamming distance retrieval and quantization error in cosine similarity. In Section 3.2, we propose to maximize cosine similarity between the continuous codes and binary orthogonal target under a single classification objective (for both single-label and multi-labels classification).

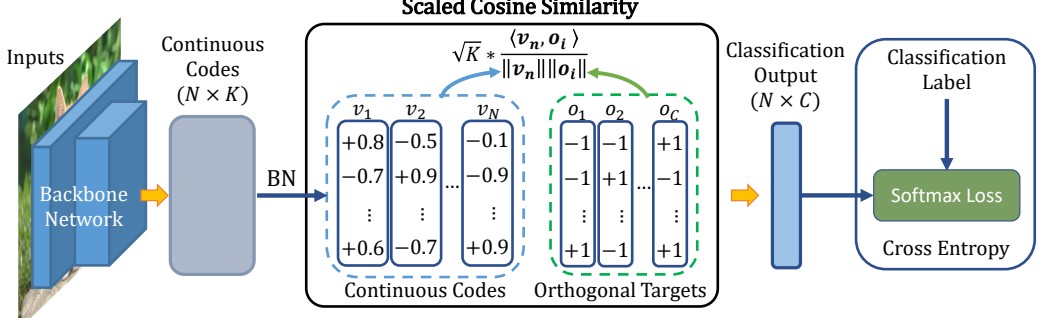

Figure 2: We first obtain continuous codes $\mathbf{V} = \{\mathbf{v}_n\}_{n=1}^N \in \mathbb{R}^{N \times K}$ from our backbone network. It is then passed through a batch normalization (BN) layer to obtain zero-mean continuous codes. Next, we compute scaled cosine similarity between the continuous codes and their binary orthogonal targets $\mathbf{O} = \{\mathbf{o}_i\}_{i=1}^C \in \{-1, +1\}^{C \times K}$ where C = number of classes. Finally, the scaled cosine similarity will act as a classification output and we minimize a cross entropy loss. See Section 3.2 for details.

Finally, we describe why adding a batch normalization layer after the continuous codes will achieve code balance in Section 3.2.3. Our method is illustrated in Figure 2.

Let us first formally define the deep hashing problem. Let $d$-dimensional data, $\mathbf{X} = \{\mathbf{x}_n\}_{n=1}^N \in \mathbb{R}^{N \times d}$ where $N$ is the number of training samples, and $\mathbf{Y} = \{\mathbf{y}_n\}_{n=1}^N \in \{0, 1\}^{N \times C}$ as one-hot training labels of $C$ classes (for multi-labels, $\mathbf{y}_n \triangleq \mathbf{y}_{ni} = [y_{n1}, \cdots, y_{nC}]$, whose $y_{ni} = 1$ if any $i$-th class are assigned to the $n$-th sample and $0$ otherwise). Our objective is to learn a set of $K$-bit binary codes $\mathbf{B} = \{\mathbf{b}_n\}_{n=1}^N \in \{-1, 1\}^{N \times K}$ for each training point $\mathbf{x}_n$, which is converted from the continuous codes $\mathbf{v}_n$ through a *sgn* function. $\mathbf{v}$ can be computed by a latent layer $\mathcal{H}(\mathbf{x}) = \mathbf{W}\phi(\mathbf{x}) \in \mathbb{R}^K$, $\phi(\cdot)$ is a deep neural network (backbone network) to compute $q$-dimensional nonlinear feature representation $\mathbf{f} = \phi(\mathbf{x}) \in \mathbb{R}^q$, $\mathbf{W} \in \mathbb{R}^{K \times q}$ is the weights of the latent layer and $sgn(\mathbf{v}_{nk}) = 1$ if $k$-th bit of $\mathbf{v}_n \geq 0$ and $-1$ otherwise. In our work, binary orthogonal targets $\mathbf{o}_{y_n} \in [\mathbf{o}_1, \cdots, \mathbf{o}_C]^\mathsf{T} = \mathbf{O} \in \{-1, +1\}^{C \times K}$, where $\mathbf{o}_i$ denotes a column vector belongs to $i$-th class. Ideally, for any two rows, $1 \leq i, j \leq C$, $\mathbf{o}_i$ and $\mathbf{o}_j$ are orthogonal to each other. We use $a$ or $A$ to represent scalar, $\mathbf{a}$ to represent column vector, and $\mathbf{A}$ to represent matrix. Both $i$ and $j$ are often used as index.

## 3.1 Reformulating Deep Hashing in the Lens of Cosine Similarity

**Interpreting Hamming Distance as Cosine Similarity.** Typically, Hamming distance can be computed using logical *xor* operation between binary codes $\mathbf{b}_i$ and $\mathbf{b}_j$, followed by *popcount*. If $\mathbf{b}$ is represented by $\{-1, +1\}^K$, then Hamming distance can also be computed mathematically as:

$$D(\mathbf{b}_i, \mathbf{b}_j) = \frac{K - \mathbf{b}_i^\mathsf{T} \mathbf{b}_j}{2}. \tag{1}$$

Geometrically, the dot product $\mathbf{b}_i^\mathsf{T} \mathbf{b}_j$ can be interpreted as:

$$\mathbf{b}_i^\mathsf{T} \mathbf{b}_j = \|\mathbf{b}_i\| \, \|\mathbf{b}_j\| \cos\theta_{ij}, \tag{2}$$

in which $\|\cdot\|$ is the Euclidean norm and $\theta_{ij}$ is the angle between $\mathbf{b}_i$ and $\mathbf{b}_j$. As both $\|\mathbf{b}_i\|$ and $\|\mathbf{b}_j\|$ are constant (i.e., $\|\mathbf{b}\| = \sqrt{K}$), equation (1) can then be viewed as:

$$D(\mathbf{b}_i, \mathbf{b}_j) = \frac{K - K\cos\theta_{ij}}{2} = \frac{K}{2}(1 - \cos\theta_{ij}). \tag{3}$$

Since $\frac{K}{2}$ is a constant, we can see that the retrieval is now will be only based on the angle between two hash codes i.e., similar hash codes will have a similar direction, yield a lower angle between them, and hence a lower hamming distance.

**Interpreting Quantization Error as Cosine Similarity.** Typically, converting continuous codes $\mathbf{v}$ to binary codes $\mathbf{b}$ will lead to information loss, which is also known as quantization error. Therefore,

most of the existing hashing methods have included quantization error minimization in their learning objective such as $L_1$-norm, $L_2$-norm and p-norm (e.g., $p = 3$ in Greedy Hash [41]), usually in the form of:

$$\min L + \lambda Q, \qquad (4)$$

where $L$ is the supervised learning objective such as Cross Entropy and $Q$ is the quantization error between $\mathbf{v}$ and $\mathbf{b}$. However, it is difficult to control the scale $\lambda$, i.e. a low $\lambda$ might not be effective, while a high $\lambda$ might lead to underfitting. As a result of this, careful tuning is needed and yet the tuned $\lambda$ may varies in different tasks. To overcome this cumbersome practise, let us first interpret quantization error geometrically:

$$\min \|\mathbf{v} - \mathbf{b}\|^2 \quad \text{s.t.} \quad \mathbf{b} \in \{-1, 1\}^K, \qquad (5)$$

in which $\mathbf{v}$ is in continuous space, $\mathbf{b} = sgn(\mathbf{v})$ is in binary space. We expand equation (5) to get:

$$\|\mathbf{v} - \mathbf{b}\|^2 = \|\mathbf{v}\|^2 + \|\mathbf{b}\|^2 - 2\|\mathbf{v}\|\|\mathbf{b}\|\cos\theta_{vb}. \qquad (6)$$

According to equation (3), retrieval is only based on the similarity in the direction of two hash codes. Hence, we can ignore the magnitude of $\mathbf{v}$ by normalizing it to have the same norm with $\mathbf{b}$, i.e., $\|\mathbf{v}\| = \sqrt{K}$ and interpret the quantization error as to only the angle $\theta_{vb}$ between $\mathbf{v}$ and $\mathbf{b}$[3]:

$$\|\mathbf{v} - \mathbf{b}\|^2 = 2K - 2K\cos\theta_{vb} = 2K(1 - \cos\theta_{vb}). \qquad (7)$$

Since $2K$ is a constant, we can then conclude that maximize the cosine similarity between $\mathbf{v}$ and $\mathbf{b}$ will lead to a low quantization error, leading to a better approximation in the hash codes.

## 3.2 Discriminative Hash Codes with Orthogonal Target

According to [6], the probability of two samples $\mathbf{x}_i$ and $\mathbf{x}_j$ to have the same hash code under a family $\mathcal{F}$ of hash functions using random hyperplane technique can be described as $\mathbf{Pr}_{h \in \mathcal{F}}[h(\mathbf{x}_i) = h(\mathbf{x}_j)] = 1 - \frac{\theta_{ij}}{\pi}$, where $h(\cdot)$ is a hash function and $\theta_{ij}$ is the angle between $\mathbf{x}_i$ and $\mathbf{x}_j$. Therefore, based on the same principle, it can be derived that if the two continuous codes $\mathbf{v}_i$ and $\mathbf{v}_j$ from latent layer $\mathcal{H}$ have high cosine similarity, then the hash codes $\mathbf{b}_i$ and $\mathbf{b}_j$ should also have high chance of obtaining the same hash codes. Beside that, as described in Section 3.1, cosine similarity can also be used to justify the retrieval performance using both the hash codes and quantization error between the continuous codes and hash codes. Given these two circumstances, we therefore propose to maximize the cosine similarity of the continuous codes $\mathbf{v}_n$ and its corresponding **binary orthogonal target**, $\mathbf{o}_{y_n} \in [\mathbf{o}_1, \cdots, \mathbf{o}_C]^\intercal = \mathbf{O} \in \{-1, +1\}^{C \times K}$, where this can be achieved by maximizing the posterior probability of the ground-truth class using softmax (cross-entropy) loss:

$$L = -\frac{1}{N} \sum_{n=1}^{N} \log \frac{\exp\left(\mathbf{o}_{y_n}^\intercal \mathbf{v}_n\right)}{\sum_{i=1}^{C} \exp\left(\mathbf{o}_i^\intercal \mathbf{v}_n\right)}, \qquad (8)$$

where $\mathbf{v}_n$ denotes the deep continuous codes of the $n$-th samples from DNN $\phi$ and both $\mathbf{o}_{y_n}$, $\mathbf{o}_i \in \mathbf{O}$ denote the ground-truth class $y_n$ and the $i$-th class of the binary orthogonal targets. For simplicity, we omit the bias term from equation (8). It follows that under the framework of deep hypersphere embedding [35, 42, 10], we can transform the logit $\mathbf{o}_i^\intercal \mathbf{v}_n = \|\mathbf{o}_i\| \|\mathbf{v}_n\| \cos\theta_{ni}$ where $\theta_{ni}$ is the angle between the continuous codes $\mathbf{v}_n$ and the binary orthogonal target $\mathbf{o}_i$. Next, we perform $L_2$ normalization on $\mathbf{v}_n$ so that $\|\mathbf{v}_n\| = 1$, and $\|\mathbf{o}_i\| = \sqrt{K}$ since it is in binary form. Now our loss function can be rewritten as:

$$L = -\frac{1}{N} \sum_{n=1}^{N} \log \frac{\exp\left(\sqrt{K}\cos\theta_{y_n}\right)}{\exp\left(\sqrt{K}\cos\theta_{y_n}\right) + \sum_{i=1, i \neq y_n}^{C} \exp\left(\sqrt{K}\cos\theta_{ni}\right)}. \qquad (9)$$

As such, instead of introducing the quantization error minimization in the learning objective (equation (4)), our proposed method unifies both the learning objective and quantization error minimization together under *a single classification objective* as shown in the loss function (equation (9)). Furthermore, since the binary orthogonal targets attain maximal inter-class Hamming distance and that our loss function also aims to minimize the intra-class variance, we can leverage on cosine or angular

---

[3]See Appendix B in supplementary material for proof.

margin[4] that have been proven to be beneficial in CosFace [42] and ArcFace [10], to further improve the minimization of intra-class variance (we set $m = 0.2$ in all of our experiments unless mentioned explicitly). With this, our method is able to perform end-to-end training to learn highly discriminative hash codes without both the sophisticated training objectives and computational graph modifications.

### 3.2.1 Binary Orthogonal Target

The maximization of the expectation of inter-class Hamming distance will help to increase the recall rate during retrieval as there will be lesser chance to retrieve incorrect items, because the aim is to retrieve more similar items (intra-class), and avoid to retrieve incorrect items (inter-class). That is, given a K-bit Hamming space $\mathbb{H}^K \in \{-1, +1\}^K$, for any two binary vectors $\mathbf{b}_i, \mathbf{b}_j$ sampled with probability $p$ for $+1$ on each bit, the expectation of Hamming distance is $\mathbb{E}[D(\mathbf{b}_i, \mathbf{b}_j)] = 2 \cdot K \cdot p(1-p)$ and it achieves the upper bound of $\frac{K}{2}$ with $p = 0.5$ [12, 50] (See Appendix B in supplementary material for details.). Hence, hash codes $\mathbf{b}_i$ and $\mathbf{b}_j$ must be orthogonal so that we can get $D(\mathbf{b}_i, \mathbf{b}_j) = \frac{K}{2}$ in equation (3).

**Orthogonal Targets Generation.** Hadamard matrix naturally contains orthogonal rows and columns, which guarantees the maximum Hamming distance of $\frac{K}{2}$ between any two rows [50, 29]. However, it is restricted when $K$ is not 1, 2, or a multiple of 4. Hence, a simple solution is to sample the targets from $Bern(0.5)$ which every sampled bit has the probability $p = 0.5$ to be $+1$. The result is the expectation of Hamming distance between any two rows equals to $\frac{K}{2}$ which indicates orthogonality. One limitation is that if $2^K < C$, the nearest rows in the sampled targets will be identical, which causes performance degrade. Hence, a simple solution is to increase $K$. In supplementary material (Appendix D.3), we show that the two nearest rows has Hamming distance closed to $\frac{K}{2}$ as $K$ is higher. We also generate the targets with the objective of maximum inter-class Hamming distance heuristically, it indeed improved the performance at lower $K$, but the improvement in higher $K$ are negligible.

### 3.2.2 Multi-labels Hash Codes Learning

As conventional cross-entropy loss only works for single-label classification, we leverage the concept of Label Smoothing [38] to generate labels for multi-labels classification. A standard cross entropy (CE) loss is mathematically formulated as:

$$L = -\frac{1}{N} \sum_{n=1}^{N} \sum_{i=1}^{C} y_{ni} \log(p(y_{ni}|\mathbf{x}_n)), \tag{10}$$

in which $y_{ni} = 1$ if $i$-th class is assigned to the $n$-th sample in a single label multiclass classification task. In [38], the target label becomes soft-target such that non-target class has a small "smoothing" value to regularize overconfident samples and we leverage this concept for multi-labels. To adopt CE for multi labels classification, we set $y_{ni} = z > 0$ if any $i$-th class are assigned to $n$-th sample. The constant $z$ is determined such that $\sum_{i=1}^{C} y_{ni} = 1$, e.g., $z = 0.5$ and $\mathbf{y}_n = [0, 0.5, 0, 0.5]$ when the 2[nd] and the 4[th] classes are the assigned classes. Our motivation is that the model should maximize the probabilities of the target classes, which can optimize the hash codes to be as similar as the binary targets from assigned classes[5]. In our experiments, we found out empirically that replacing *softmax* with *sigmoid* for multi-labels are not effective[6]. A likely explanation is that *softmax* will intrinsically suppress the lower activated class unit (i.e., scaled cosine similarity) with lower probability and increase the highly activated class unit with higher probability, while *sigmoid* will treat each class unit as an individual unit. As a result, maximizing probability of a class might not lead to minimizing the probability of other classes. Therefore, we propose to leverage the concept of Label Smoothing to generate labels so that we can use cross entropy loss for learning.

---

[4]Cosine margin will transform $\exp(\sqrt{K}\cos\theta_{y_n})$ to $\exp(\sqrt{K}(\cos(\theta_{y_n}) - m))$ and angular margin will transform the same to $\exp(\sqrt{K}\cos(\theta_{y_n} + m))$.

[5]Note, we cannot guarantee that the final hash codes are the center of hash codes of the target classes. Instead we let the optimization algorithm to find the best hash codes.

[6]See Appendix D.4 in supplementary material for details.

### 3.2.3 Code Balance

Although binary orthogonal target helps in code balancing, since every bit has 50% of chance being $+1$ or $-1$, there is no guarantee that the model will learn to output a balanced code. Therefore, we propose to add a batch normalization (BN) layer after the continuous codes $v$ to ensure the code balance. If $\sum_n \mathbf{v}_{nk} = 0$, then we can see that $\sum_n \mathbf{b}_{nk} = 0$ for the $k$-th bit. Because the distribution of $\mathbf{v}$ has been normalized to have zero-mean and variance of 1, with $\mathbf{b} = sgn(\mathbf{v})$, the hash codes $\mathbf{b}$ will follow a uniform binary distribution with 50% chances on both $+1$ and $-1$. Empirically, we found that it improves the retrieval performance on ImageNet100 by about 17-20% as compared with a model with normal cross entropy loss (see Table 1). Note that the Bi-half method [27] shifts the continuous codes by their median, followed by converting the continuous codes to binary codes for optimization. However, it will have to modify the computational graph in order to have a proxy derivative to the solve vanishing gradient problem. In contrast, appending BN layer will not modify the computational graph, therefore enabling straightforward end-to-end training.

## 4 Experiment

**Training Setup**. We select 7 different deep hashing methods for comparison (5 point-wise, 1 pair-wise and 1 triplet-wise). For a fair comparison, we use the same learning rate of $0.0001$, *Adam* optimizer [18] and 100 epochs for all methods. For **SDH-C** [31], we have modified it from pair-wise objective to point-wise objective, while all penalty terms are kept (i.e., quantization loss, bit variance loss and orthogonality on projection weights).

**Datasets**. We follow prior works [5, 12, 41, 33, 40, 48, 23, 34] and choose **ImageNet100** [9], **NUS-WIDE** [8] and **MS-COCO** [30] for category-level retrieval experiments. For a more practical yet challenging large-scale instance-level retrieval task (i.e., tremendous number of classes), we evaluate on the popular **GLDv2** [47], $\mathcal{R}$**Oxf** and $\mathcal{R}$**Par** [39].

**Architecture**. For category-level retrieval, following the settings in [5, 41, 40, 12], we use pre-trained AlexNet [20] as the network backbone initialization. The output from last fully-connected with ReLU (4096-dimension vector) acts as input to the latent layer; various supervised deep hashing methods are then applied to generate binary codes. The image size is $224 \times 224$. For instance-level retrieval, due to the expensive cost of training from scratch, we use pre-trained model[7] (**R50-DELG-GLDv2-clean**) from DELG [3] to compute the 2048-dimension global descriptors. We then train a latent layer $\mathcal{H}$ to compute hash codes where inputs are the global descriptors. For GLDv2, the images input are $512 \times 512$. For $\mathcal{R}$Oxf and $\mathcal{R}$Par, we use 3 scales $\{\frac{1}{\sqrt{2}}, 1, \sqrt{2}\}$ to produce multi-scale representations. These are subject to $\text{L}_2$ normalization, and then average-pooled to obtain a single descriptor as done by [3]. A *GLDv2-trained* latent layer is used to compute hash codes for the evaluations.

Details of training setups, datasets and architecture can be found in the supplementary material (Appendix C).

### 4.1 Results on Category-level Retrieval

For performance evaluation[8], we use mean average precision (mAP@R) which is the mean of average precision scores of the top R retrieved items. Table 1 offers performance comparison amongst all selected hashing methods and our methods (+variants). **CE** denotes model trained with cross entropy only, the hash codes are computed from sign of continuous codes. **CE+BN** denotes **CE** model with BN layer [17] appended after the latent layer. **CE+Bihalf** denotes **CE** model with Bihalf[9] [27] layer appended after the latent layer. **OrthoCos** denotes model trained with cosine margin and binary orthogonal target. **OrthoCos+Bihalf** denotes a variant of **OrthoCos**, and with Bihalf layer appended. **OrthoCos+BN** denotes a variant of **OrthoCos**, and with BN layer appended. **OrthoArc+BN** denotes a variant of **OrthoCos+BN**, trained with angular margin.

**Overall.** It can be observed that both our **OrthoCos+BN** and **OrthoArc+BN** perform better than recent state-of-the-art, **DPN** [12] and **CSQ** [50]. On multi-labeled datasets (i.e., NUS-WIDE and MS

---

[7]https://github.com/tensorflow/models/tree/master/research/delf

[8]See Appendix C.3 in supplementary material for evaluation detail.

[9]Note that Bihalf was originally proposed for unsupervised hashing, however, we think it is worth to compare as Bihalf is a layer for code balancing instead of a training objective.

| Methods | ImageNet100 (mAP@1K) | | | | NUS-WIDE (mAP@5K) | | | | MS COCO (mAP@5K) | | | |
|---|---|---|---|---|---|---|---|---|---|---|---|---|
| | 16 | 32 | 64 | 128 | 16 | 32 | 64 | 128 | 16 | 32 | 64 | 128 |
| HashNet[2] [5] | 0.343 | 0.480 | 0.573 | 0.612 | 0.814 | 0.831 | 0.842 | 0.847 | 0.663 | 0.693 | 0.713 | 0.727 |
| DTSH[3] [45] | 0.442 | 0.528 | 0.581 | 0.612 | **0.816** | **0.836** | **0.851** | **0.862** | 0.699 | 0.732 | 0.753 | 0.770 |
| SDH-C[1] [31] | 0.584 | 0.649 | 0.664 | 0.662 | 0.763 | 0.792 | 0.816 | 0.832 | 0.671 | 0.710 | 0.733 | 0.742 |
| GreedyHash[1] [41] | 0.570 | 0.639 | 0.659 | 0.659 | 0.771 | 0.797 | 0.815 | 0.832 | 0.677 | 0.722 | 0.740 | 0.746 |
| JMLH[1] [40] | 0.517 | 0.621 | 0.662 | 0.678 | 0.791 | 0.825 | 0.836 | 0.843 | 0.689 | 0.733 | 0.758 | 0.768 |
| DPN[1] [12] | 0.592 | 0.670 | 0.703 | 0.714 | 0.783 | 0.818 | 0.838 | 0.842 | 0.668 | 0.721 | 0.752 | 0.773 |
| CSQ[1] [50] | 0.586 | 0.666 | 0.693 | 0.700 | 0.797 | 0.824 | 0.835 | 0.839 | 0.693 | **0.762** | 0.781 | 0.789 |
| CE[1] | 0.350 | 0.379 | 0.406 | 0.445 | 0.744 | 0.770 | 0.796 | 0.813 | 0.602 | 0.639 | 0.658 | 0.676 |
| CE+BN[1] | 0.533 | 0.586 | 0.612 | 0.617 | 0.801 | 0.814 | 0.823 | 0.825 | 0.697 | 0.721 | 0.729 | 0.726 |
| CE+Bihalf[1] [27] | 0.541 | 0.630 | 0.661 | 0.662 | 0.802 | 0.825 | 0.836 | 0.839 | 0.674 | 0.728 | 0.755 | 0.757 |
| OrthoCos[1] | 0.583 | 0.660 | 0.702 | 0.714 | 0.795 | 0.826 | 0.842 | 0.851 | 0.690 | 0.745 | 0.772 | 0.784 |
| OrthoCos+Bihalf[1] | 0.562 | 0.656 | 0.698 | 0.711 | 0.804 | 0.834 | 0.846 | 0.852 | 0.690 | 0.746 | 0.775 | 0.782 |
| OrthoCos+BN[1] | 0.606 | 0.679 | **0.711** | **0.717** | 0.804 | **0.836** | 0.850 | 0.856 | **0.709** | 0.762 | **0.787** | **0.797** |
| OrthoArc+BN[1] | **0.614** | **0.681** | 0.709 | 0.714 | 0.806 | 0.833 | 0.850 | 0.856 | 0.708 | **0.762** | 0.785 | 0.794 |

Table 1: Performance of different methods for 4 different bits on different benchmark datasets. All results are run by us. The superscript [1], [2] and [3] indicate point-wise, pair-wise and triplet-wise method respectively. **Bold** values indicate best performance in the column.

| Methods | GLDv2 (mAP@100) | | | $\mathcal{R}$Oxf-Hard (mAP@all) | | | $\mathcal{R}$Paris-Hard (mAP@all) | | |
|---|---|---|---|---|---|---|---|---|---|
| | 128 | 512 | 2048 | 128 | 512 | 2048 | 128 | 512 | 2048 |
| HashNet[2] [5] | 0.018 | 0.069 | 0.111 | 0.034 | 0.058 | 0.307 | 0.133 | 0.190 | 0.490 |
| DPN[1] [12] | 0.021 | 0.089 | 0.133 | 0.053 | 0.184 | 0.303 | 0.224 | 0.399 | 0.562 |
| GreedyHash[1] [41] | 0.029 | 0.108 | 0.144 | 0.032 | 0.251 | 0.373 | 0.128 | 0.531 | 0.652 |
| CSQ[1] [50] | 0.023 | 0.086 | 0.114 | 0.093 | 0.284 | 0.398 | 0.245 | 0.541 | 0.649 |
| OrthoCos+BN[1] | **0.035** | **0.111** | **0.147** | **0.184** | **0.359** | **0.447** | **0.416** | **0.608** | **0.669** |
| R50-DELG-H | - | - | 0.125* | - | - | 0.471 | - | - | 0.682 |
| R50-DELG-C | - | - | 0.138* | - | - | 0.510 | - | - | 0.715 |

Table 2: Performance of different methods for 3 different numbers of bits on different instance-level benchmark datasets. All results are run by us. The superscript [1] and [2] indicate point-wise and pair-wise method respectively. **Bold** values indicate best performance in the column. * indicates using $512 \times 512$ image inputs, hence different performance as reported by DELG [3]. R50-DELG-H denotes Hamming distance retrieval using the sign of extracted descriptors. R50-DELG-C denotes Cosine distance retrieval using the extracted descriptors.

COCO), **DTSH** [45] (triplet based method) performed the best with 0.851 and 0.862 with 64 and 128-bits hash codes in NUS-WIDE followed by our method (e.g., **OrthoCos+BN** achieves 0.850 and 0.856 in the same settings), while **OrthoCos+BN** and **OrthoArc+BN** performed the best on MS-COCO with at most 1% improvement over previous deep hashing methods.

**Code Balance.** Although retrieval performance of **CE** models performed the worst, but by appending BN layer after the latent layer (**CE+BN**), we were able to observe 5-20% improvement over all settings (dataset and number of bits). Bihalf [27] layer (zero-median features) has a proxy derivative to learn hash features, hence getting 0.1-4.9% improvement than **CE+BN**. This indicates that without sophisticated training objectives, code balance itself is a very important factor in improving Hamming distance based retrieval. However, **OrthoCos+Bihalf** does not show significant improvement over **OrthoCos+BN**, but is comparable with **OrthoCos**. We thus conclude that our method can achieve code balance without explicitly engineering the computational graph.

**Cosine and Angular Margin.** In our experiments, we observed that cosine margin (**OrthoCos+BN**) slightly outperform angular margin (**OrthoArc+BN**) by about 0.2% on average.

### 4.2 Results on Instance-Level Retrieval

For evaluation metrics, we adapt the evaluation protocol of [3, 39]. The baseline performance of GLDv2, $\mathcal{R}$Oxf-Hard and $\mathcal{R}$Par-Hard from the pre-trained **R50-DELG-GLDv2-clean** are 0.138,

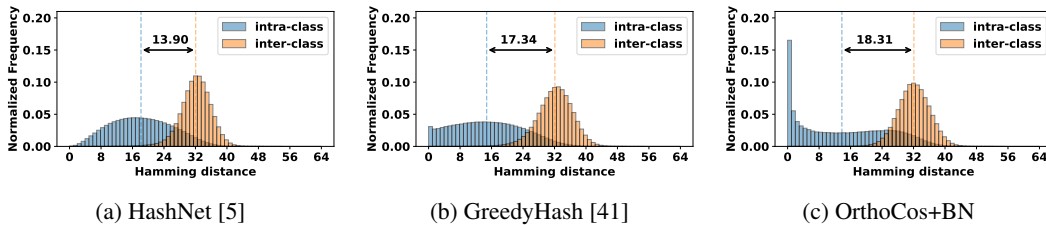

| (a) HashNet [5] | (b) GreedyHash [41] | (c) OrthoCos+BN |

Figure 3: Histogram of intra-class and inter-class Hamming distances with 64-bits ImageNet100. The arrow annotation is the separability in Hamming distances, $\mathbb{E}[D_{inter}] - \mathbb{E}[D_{intra}]$. We normalized the frequency so that sum of all bins equal to 1.

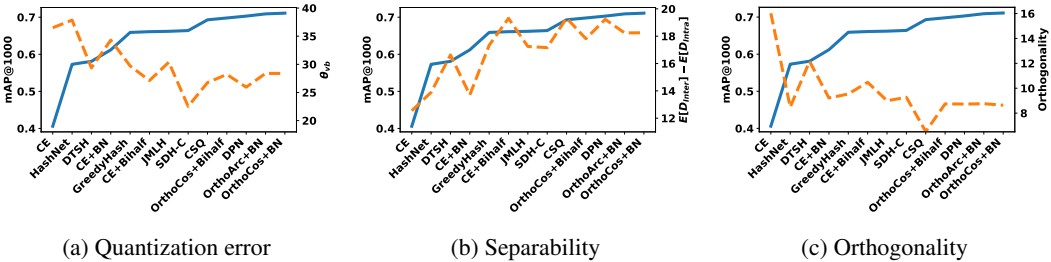

| (a) Quantization error | (b) Separability | (c) Orthogonality |

Figure 4: Analysis of retrieval performance of 64-bits ImageNet100. **(a)** Quantization error: $\theta_{vb}$. **(b)** Separability: $\mathbb{E}[D_{inter}] - \mathbb{E}[D_{intra}]$. **(c)** Orthogonality: $\left\| \frac{1}{K} \mathbf{H}\mathbf{H}^\intercal - \mathbb{I} \right\|$. Blue solid line denotes mean average precision (mAP@1000) and orange dotted line denotes the respective analysis score.

0.510 and 0.715 respectively. Table 2 summarizes the performance of different deep hashing methods and our method. For all the 3 datasets, our method outperforms all previous deep hashing methods on all bits. This suggests that our method has a better generalization ability on unseen instances than previous deep hashing methods. In particular, our model significantly outperforms previous deep hashing models by 0.6%, 9.1% and 17.1% respectively on the 3 datasets with 128-bits hash codes.

**Orthogonal Transformation.** For GLDv2 2048-bits hash codes, surprisingly it can achieve a much better performance than the pre-trained 2048-dimensions descriptors (by 1.1% improvement over R50-DELG-C). We then analyze the separability in cosine distances, i.e., the difference in the mean of intra-class cosine distance and the mean of inter-class cosine distance before and after the transformation (similar to Figure 3). We observe that the separability in cosine distances increases after the orthogonal transformation, i.e., before it is 0.142 and after it increases to 0.167. The results thus show that learning orthogonal hash codes can transform the inputs to be more discriminative.

**Domain shifting with BN.** As the model is trained with GLDv2, the running mean and variance in the BN layer might experience domain shifting problem [26] when testing directly on different datasets (e.g., $\mathcal{R}$Oxf and $\mathcal{R}$Par). We empirically found that using running mean and variance from GLDv2 will lead to a large performance drop in Hamming distance retrieval[10]. One simple solution is to recompute the mean and variance from all continuous codes in the database, then update the running mean and variance with the computed mean and variance. The performances of $\mathcal{R}$Oxf and $\mathcal{R}$Par in Table 2 are obtained with running mean and variance of the respective database.

### 4.3 Further Analysis

**Histogram of Hamming Distances.** Figure 3 summarizes the histogram of intra-class and inter-class distances. We compare our method **OrthoCos+BN** with pair-wise method HashNet [5] and point-wise classification based GreedyHash [41]. Although the distribution of inter-class distances are about the same for all the 3 methods (close to Hamming distance of $K/2 = 32$), we can see that the larger the separability i.e., the difference in the mean of intra-class distance (the blue dotted line) with the mean of inter-class distance (the orange dotted line), the better the performance.

---

[10]See Appendix D.5 in supplementary material for details.

**Performance Improvement Analysis.** We further analyze the reasons behind performance improvements of different deep hashing methods, and summarizes the results in Figure 4. We conclude 3 main reasons that contribute to the improvement in deep hashing methods: i) quantization error; ii) the separability in Hamming distances; and iii) orthogonality in hash centers. For quantization error, we measure the angle $\theta_{vb}$ between the continuous codes $\mathbf{v}$ and the hash codes $\mathbf{b}$, i.e., $\theta_{vb} = deg(\arccos(\frac{\mathbf{v}^\intercal \mathbf{b}}{\|\mathbf{v}\|\|\mathbf{b}\|}))$. For separability, we measure the difference in the mean of inter-class distances and the mean of intra-class distances, i.e., $\mathbb{E}[D_{inter}] - \mathbb{E}[D_{intra}]$. For orthogonality, we first compute the hash centers $\mathbf{H} \in \{-1, +1\}^{C \times K}$ for every class (by taking the sign of average hash codes in every class), then we measure the orthogonality with $\left\| \frac{1}{K}\mathbf{H}\mathbf{H}^\intercal - \mathbb{I} \right\|$ (lower is better). When the quantization error reduces, the separability increases and the hash centers has better orthogonality, resulting in better performance.

## 5 Conclusion & Future Work

We propose to unify training objectives of deep hashing under a single classification objective. We show this can be achieved by maximizing the cosine similarity between the continuous codes and binary orthogonal target under a cross entropy loss. For that, we first reformulated the problem of deep hashing in the lens of cosine similarity. We then demonstrated that if we perform $L_2$-normalization on the continuous codes, then end-to-end training of deep hashing is possible without any extra sophisticated constraints. Moreover, we leverage the concept of Label Smoothing to train multi-labels classification with cross-entropy loss and batch normalization for code balancing. Extensive experiments validated the efficiency of our method in both category-level and instance-level retrieval benchmarks.

Nonetheless, the proposed method might fail when the number of bits is too small (<8 bits), especially when number of classes is much greater than the number of bits. In this case, there will be overlapping in the generated target code (i.e., the number of maximum unique codes is equal to $2^K$ where $K$ is number of bits). In such condition, the target code will also not guarantee to be orthogonal. Overcoming this limitation is part of the future work. Also, we are exploring how to learn better feature representations to improve the retrieval performance by using hash codes through unsupervised learning.

## Broader Impact

Hashing remains a key bottleneck in practical deployments of large-scale retrieval systems. Recent deep hashing frameworks have shown great promise in learning code that are both compact and discriminative. Yet state-of-the-art frameworks are known to be difficult to train and to reproduce – largely owing to their complex loss designs that dictates hyperparameter tuning and multi-stage training. In this work, we set out to change that – we attempt to unify deep hashing under *a single objective*, therefore simplifying training and help reproducibility. Our key intuition lies with reformulating hashing in the lens of cosine similarity. We report competitive hashing performance on all common datasets, and significant improvements over state-of-the-arts on the more challenging task of instance-level retrieval.

## Acknowledgments and Disclosure of Funding

This research is partly supported by the Fundamental Research Grant Scheme (FRGS) MoHE Grant FP021-2018A, from the Ministry of Education Malaysia. We also thank Kilho Shin for helpful discussions and recommendations.

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
