# Supplementary Material for One Loss for All: Deep Hashing with a Single Cosine Similarity based Learning Objective

**Jiun Tian Hoe**[1*]   **Kam Woh Ng**[2,3*]   **Tianyu Zhang**[4]
**Chee Seng Chan**[1†]   **Yi-Zhe Song**[2,3]   **Tao Xiang**[2,3]

[1]CISiP, Universiti Malaya, Malaysia
[2]CVSSP, University of Surrey, U.K.
[3]iFlyTek-Surrey Joint Research Centre on Artificial Intelligence
[4]Geek+, China

## A   Summary

In Appendix B, we explain in details the proof in the main paper. In Appendix C, we describe in details all the training setups, hyper-parameters, datasets and evaluation details. In Appendix D, we performed more experiments on ablation study and further analysis. We will release the code upon publications.

## B   Proof

**Interpreting Quantization Error as Cosine Similarity.** Quantization error between the continuous codes $\mathbf{v}$ and the hash codes $\mathbf{b}$ can be interpreted geometrically:

$$\min \|\mathbf{v} - \mathbf{b}\|^2 \quad \text{s.t.} \quad \mathbf{b} \in \{-1, 1\}^K, \tag{1}$$

in which $\mathbf{v}$ is in continuous space, $\mathbf{b} = sgn(\mathbf{v})$ is in binary space. We expand equation (1) to get:

$$\begin{aligned}
\|\mathbf{v} - \mathbf{b}\|^2 &= \langle \mathbf{v} - \mathbf{b} , \mathbf{v} - \mathbf{b} \rangle \\
&= \|\mathbf{v}\|^2 - \langle \mathbf{v} , \mathbf{b} \rangle - \langle \mathbf{b} , \mathbf{v} \rangle + \|\mathbf{b}\|^2 \\
&= \|\mathbf{v}\|^2 - 2\langle \mathbf{v} , \mathbf{b} \rangle + \|\mathbf{b}\|^2 \\
&= \|\mathbf{v}\|^2 + \|\mathbf{b}\|^2 - 2 \|\mathbf{v}\| \|\mathbf{b}\| \cos \theta_{vb}.
\end{aligned} \tag{2}$$

According to equation (3) in the main paper, retrieval is only based on the similarity in the direction of two hash codes. Hence, we can ignore the magnitude of $\mathbf{v}$ by normalizing it to have the same norm with $\mathbf{b}$, i.e., $\|\mathbf{v}\| = \sqrt{K}$ and interpret quantization error as only the angle $\theta_{vb}$ between $\mathbf{v}$ and $\mathbf{b}$:

$$\begin{aligned}
\|\mathbf{v} - \mathbf{b}\|^2 &= \|\mathbf{v}\|^2 + \|\mathbf{b}\|^2 - 2 \|\mathbf{v}\| \|\mathbf{b}\| \cos \theta_{vb} \\
&= \sqrt{K}^2 + \sqrt{K}^2 - 2\sqrt{K}\sqrt{K} \cos \theta_{vb} \\
&= 2K - 2K \cos \theta_{vb} = 2K(1 - \cos \theta_{vb}).
\end{aligned} \tag{3}$$

Since $2K$ is a constant, we can then conclude that maximize the cosine similarity between $\mathbf{v}$ and $\mathbf{b}$ leads to a low quantization error, leading to a better approximation in hash codes.

---

[*]equal contribution.
[†]corresponding author (*cs.chan@um.edu.my*).

35th Conference on Neural Information Processing Systems (NeurIPS 2021).

**Expectation of Hamming Distance.** Given a K-bit Hamming space $\mathbb{H}^K \in \{-1, +1\}^K$, for any two binary vectors $\mathbf{b}_i, \mathbf{b}_j$ sampled with probability $p$ for $+1$ on each bit, the expectation of Hamming distance is $\mathbb{E}[D(\mathbf{b}_i, \mathbf{b}_j)] = 2K \cdot p(1-p)$. For any two bits, $\mathbf{b}_i, \mathbf{b}_j$, the probability of obtaining different bits:

$$
\begin{aligned}
Pr[\mathbf{b}_i \neq \mathbf{b}_j] &= Pr[(\mathbf{b}_i = +1) \wedge (\mathbf{b}_j = -1)] + Pr[(\mathbf{b}_i = -1) \wedge (\mathbf{b}_j = +1)] \\
&= p(1-p) + (1-p)p \\
&= 2p(1-p).
\end{aligned}
\tag{4}
$$

Then, the expectation of Hamming distance of 1-bit can be computed as:

$$
\begin{aligned}
\mathbb{E}[D(\mathbf{b}_i, \mathbf{b}_j)] &= 1 \cdot Pr[\mathbf{b}_i \neq \mathbf{b}_j] + 0 \cdot Pr[\mathbf{b}_i = \mathbf{b}_j] \\
&= 1 \cdot Pr[\mathbf{b}_i \neq \mathbf{b}_j] + 0 \cdot (1 - Pr[\mathbf{b}_i \neq \mathbf{b}_j]) \\
&= Pr[\mathbf{b}_i \neq \mathbf{b}_j] \\
&= 2p(1-p).
\end{aligned}
\tag{5}
$$

Since every bit in a binary code is independent sampled, hence the expectation of Hamming distance of K-bits can be computed as:

$$
\begin{aligned}
\mathbb{E}[D(\mathbf{b}_i, \mathbf{b}_j)] &= \sum_{k=1}^{K} \mathbb{E}[D(\mathbf{b}_i, \mathbf{b}_j)] \\
&= \sum_{k=1}^{K} 2p(1-p) \\
&= 2K \cdot p(1-p).
\end{aligned}
\tag{6}
$$

## C  Training Setup

**Code Implementations.** For HashNet [2], DTSH [24], GreedyHash [21], JMLH [20] and CSQ [27] methods, we referred from author's open-source repository at [3], [4], [5], [6] and [7] respectively. For SDH-C [13], DPN [6] and Bi-Half [11] methods, we implemented by ourselves according to the papers. We implemented all the methods with PyTorch [15].

**License.** The source codes of HashNet and CSQ were released under MIT license. The source code of JMLH was released under Anti 996 license. The source code of DELG [1] was released under Apache License 2.0. We didn't find any license information for the source codes of DTSH and GreedyHash.

**Architecture.** For category-level retrieval tasks (i.e., ImageNet100, NUS-WIDE, MS-COCO), we use pre-trained Alexnet [8] as the backbone for all methods, then a fully-connected layer as latent layer is appended after the outputs of the backbone (i.e., 4096-dimensions vector). Then, we set the learning rate of the backbone network to be one-tenth of the learning rate of the latent layer. For instance-level retrieval tasks (i.e., GLDv2, $\mathcal{R}$Oxf, $\mathcal{R}$Par), we use a pre-trained model released from DELG[8] (**R50-DELG-GLDv2-clean**) to compute the global descriptors (i.e., 2048-dimensions vector), then we use a fully-connected layer as the latent layer. We only train the latent layer in this setting.

**Data Augmentation.** For ImageNet100 [4] and MS-COCO [12], we perform random resized crop with crop size of $224 \times 224$ and random horizontal flips during training phase. For NUS-WIDE [3], we resize the images to $256 \times 256$ and perform random crop with crop size of $224 \times 224$ before randomly flip it in horizontal. We normalize image inputs with means of 0.485, 0.456, 0.406 and standard deviation of 0.229, 0.224, 0.225 for each channel. For GLDv2, $\mathcal{R}$Oxf and $\mathcal{R}$Par, we didn't perform any data augmentations.

---

[3]https://github.com/thuml/HashNet
[4]https://github.com/Minione/DTSH
[5]https://github.com/ssppp/GreedyHash
[6]https://github.com/ymcidence/TBH
[7]https://github.com/yuanli2333/Hadamard-Matrix-for-hashing
[8]https://github.com/tensorflow/models/tree/master/research/delf

## C.1 Hyper-parameters

| Methods | Hyperparameters |
|---|---|
| HashNet | $\alpha = 1, \beta = 1$ |
| DTSH | $\alpha = 5, \lambda = 1$ |
| SDH-C | $\alpha = 1, \lambda_0 = 0.001, \lambda_1 = 0.001, \lambda_2 = 0.001$ |
| GreedyHash | $\alpha = 1, p = 3$ |
| JMLH | $\lambda = 0.1$ |
| DPN | $m = 1$ |
| CSQ | $\lambda_1 = 0.001$ |
| CE | – |
| CE+BN | – |
| CE+BiHalf | $\gamma = 6$ |
| OrthoCos | $m = 0.2, s = \sqrt{K}$ |
| OrthoCos+BiHalf | $m = 0.2, s = \sqrt{K}, \gamma = 6$ |
| OrthoCos+BN | $m = 0.2, s = \sqrt{K}$ |
| OrthoArc+BN | $m = 0.2, s = \sqrt{K}$ |

Table 1: Hyper-parameters for methods. For **SDH-C** method, $\alpha, \lambda_0, \lambda_1, \lambda_2$ are the hyperparameters for classification objective, quantization, bit variance and orthogonality on projection weights respectively. For methods using **BiHalf**, we follow author's open source source code[9]for the hyperparameter $\gamma$. For **OrthoCos**, **OrthoCos+BiHalf**, **OrthoCos+BN** and **OrthoArc+BN**, $K$ represent number of bits in hash codes. For other methods, we follow the original symbols used in original papers.

For category-level retrieval tasks, we train for 100 epochs on all methods using *Adam* Optimizer [7] with initial learning rate of 0.0001, weight decay of 0.0005, $\beta_1 = 0.9$ and $\beta_2 = 0.999$. We lowered the learning rate to 0.00001 after 80 epochs of training. We train all methods with batch size of 256 on a single Nvidia Tesla P100 GPU. Table 1 summarises method-specific hyper-parameters for every method we ran for comparison and also our methods.

For instance-level retrieval datasets, we train for 10 epochs using Adam optimizer with initial learning rate of 0.001, weight decay of 0.0005, $\beta_1 = 0.9$ and $\beta_2 = 0.999$. The learning rate is lowered to 0.0001 and 0.00001 at epoch 4 and 7 respectively. We train all methods with batch size of 256 on a single Nvidia Tesla P100 GPU. We use the same method-specific hyper-parameters as the category-level retrieval tasks.

## C.2 Datasets

*ImageNet100* is a subset of ImageNet [4] with only 100 classes. We follow the settings from [2, 6, 21], all the validation images from 100 classes are used as query set while the remaining 128K images as database and 13K images are randomly sampled from the database for training.

*NUS-WIDE* [3] consists of 81 concepts with 269K multi-labeled images. We follow the settings from previous works, where we selected 195k images from 21 of the most frequent concepts. For each concept, we selected 100 images randomly as query set while the remaining images as database. Then 500 images per concept are sampled randomly from the database for training.

*MS-COCO* [12] is an image recognition, segmentation, and captioning dataset. We used the public released dataset from [2][10] where images with no category information have been pruned. Then, we obtain 122K images by combining the training and validation images. Finally, 5K images are sampled randomly as query set, with the remaining images as the database, then we random sample 10K images from the database for training.

*Google Landmark Datasets V2* (GLDv2) [25]. To understand the effectiveness of different hashing methods in large-scale instance-level retrieval (i.e., tremendous number of classes), we choose GLDv2 for large-scale experiments. Due to expensive cost of training from scratch, we use their released

---

[9]https://github.com/liyunqianggyn/Deep-Unsupervised-Image-Hashing
[10]https://github.com/thuml/HashNet/tree/master/pytorch/data/coco

pre-trained model[11] (**R50-DELG-GLDv2-clean**) to compute the global descriptor for 1.2M training images, 1129 queries and 762K database images. The descriptors are 2048-dimension vectors, act as input to the latent layer. All the images are scaled to $512 \times 512$. We use 1.2M training images to train the latent layer (i.e., GLDv2-trained), then use it to compute hash codes for queries and database images for evaluations.

$\mathcal{R}Oxf$ and $\mathcal{R}Par$ are revisited annotated datasets of Oxford [17] and Paris [18]. $\mathcal{R}Oxf/\mathcal{R}Par$ contains 4993/6322 database images, and a different query set for each, both with 70 images. We are also using pre-trained **R50-DELG-GLDv2-clean** to compute the global descriptors, but follow DELG[1] settings with 3 scales $\{\frac{1}{\sqrt{2}}, 1, \sqrt{2}\}$ to produce multi-scale image representations, and the 3 descriptors are first $L_2$ normalized, then average-pooled to obtain a single descriptor. Images are scaled from 1024 with 3 scales and the aspect ratio was remained. We then use the GLDv2-trained latent layer to compute hash codes for evaluations.

**License.** ImageNet and NUS-WIDE are released under Noncommercial license. For MS-COCO, the annotations are under Creative Commons Attribution 4.0 License and the use of the images must abide by the Flickr Terms of Use[12]. All train set images in GLDv2 have CC-BY licenses without the NonDerivs (ND) restriction, all index and test set images are licensed under CC-0 or Public Domain licenses and the annotations are licensed by Google under CC BY 4.0 license. For $\mathcal{R}Oxf$ and $\mathcal{R}Par$ datasets, all the use of images must respect the Flickr Terms & Condition of Use[13].

### C.3 Evaluation Detail

For ImageNet100, NUS-WIDE and MS-COCO datasets, we strictly follow evaluation protocol used by HashNet [2] (also by previous works [6, 21, 14, 20, 26, 9, 27]) to evaluate for mean average precision (mAP), see [14]. For GLDv2 dataset, we follow the evaluation protocol of DELG [1] to calculate the mAP scores, see [15]. For $\mathcal{R}Oxf$ and $\mathcal{R}Par$ datasets, we follow the evaluation protocol released by authors [19], see [16].

## D  Ablation Study & Further Analysis

### D.1  Effect of Cosine and Angular Margins

| Margin | OrthoCos | | | | OrthoArc | | | |
| | ImageNet100 (mAP@1K) | | MS-COCO (mAP@5K) | | ImageNet100 (mAP@1K) | | MS-COCO (mAP@5K) | |
| | 64 | 128 | 64 | 128 | 64 | 128 | 64 | 128 |
|---|---|---|---|---|---|---|---|---|
| m = 0.0 | 0.698 | 0.686 | 0.754 | 0.745 | 0.697 | 0.687 | 0.754 | 0.745 |
| m = 0.1 | 0.706 | 0.706 | 0.767 | 0.763 | 0.705 | 0.704 | 0.764 | 0.761 |
| m = 0.2 | 0.710 | 0.718 | 0.776 | 0.778 | 0.711 | 0.715 | 0.773 | 0.774 |
| m = 0.3 | 0.712 | 0.724 | 0.784 | 0.788 | 0.713 | 0.723 | 0.781 | 0.784 |
| m = 0.4 | **0.714** | **0.726** | 0.788 | 0.796 | **0.714** | **0.727** | 0.786 | 0.792 |
| m = 0.5 | 0.712 | **0.726** | **0.791** | **0.800** | 0.712 | **0.727** | **0.790** | **0.798** |

Table 2: The performance of different margins for 64 and 128 bits on different benchmark datasets.

In the main paper, we set $m = 0.2$ for optimization. In ablation study, we have performed experiments with various $m$ from the range of $m = 0.0$ to $m = 0.5$ to understand how the margin can help to further improve intra-class variance. Theoretically, if $m$ is too large, the performance will decrease and the model fails to converge because of the vanishing of the feature space which caused by

---

[11]https://github.com/tensorflow/models/tree/master/research/delf

[12]https://www.flickr.com/creativecommons/

[13]https://www.flickr.com/help/terms

[14]https://github.com/thuml/HashNet/blob/master/pytorch/src/test.py

[15]https://github.com/tensorflow/models/blob/master/research/delf/delf/python/datasets/google_landmarks_dataset/metrics.py

[16]https://github.com/filipradenovic/revisitop/blob/master/python/evaluate.py

the cosine constraint [23]. In Table 2, we summarizes the performance of various margins for ImageNet100 (single label) and MS-COCO (multi labels). We didn't repeat the experiments for 3 times, but we run with the same seed for all different margins under different margin types, i.e., cosine margin (**OrthoCos+BN**) and angular margin (**OrthoArc+BN**).

**Effect of margins.** When no margin is applied (i.e., $m = 0.0$), the performances of 128-bits models are lower than 64-bits models, a likely explanation is that 128-bits leads to overfitting without margin. As margin increases, all 128-bits models perform consistently better than 64-bits models. While the performances degrade in single-label ImageNet100 after $m = 0.4$, we notice that multi-labels MS-COCO did not show the sign of performance degrading. We suspect with two reasons: i) [23] suggested that $m \in [0, \frac{C}{C-1})$. MS-COCO has lower number of classes, i.e., $C = 80$ while ImageNet100 has $C = 100$. Hence, MS-COCO can endure with higher margin; ii) This improvement is from the regularization of label smoothing [16], which regularizes the extreme margin effect (e.g. sensitive to noisy data) and remedy the cosine constraint (e.g. the model is trained to maximize probabilities of *multiple* classes). We further running with $m > 0.5$, the performance are degrading for ImageNet100 while negligible improvement (less than 0.1%) for MS-COCO. Hence we report only the performance with $m \leq 0.5$.

**Margin Types.** We observed that cosine margin (**OrthoCos+BN**) slightly outperform angular margin (**OrthoArc+BN**) by about 0.13% on average. Hence we can conclude that using both margin methods will lead to comparable performance, both methods will improve the minimization of intra-class variance, which lead to better performance. Nevertheless, we think that cosine margin has a better benefit over the computation complexity (angular margin requires a few more computation steps than cosine margin).

### D.2  Effect of Scales in Cosine Similarity

| Scale | OrthoCos | | | | OrthoArc | | | |
| --- | --- | --- | --- | --- | --- | --- | --- | --- |
| | ImageNet100 (mAP@1K) | | MS-COCO (mAP@5K) | | ImageNet100 (mAP@1K) | | MS-COCO (mAP@5K) | |
| | 64 | 128 | 64 | 128 | 64 | 128 | 64 | 128 |
| s = 1 | 0.706 | 0.712 | 0.785 | 0.797 | 0.700 | 0.705 | 0.785 | 0.795 |
| s = 2 | 0.706 | 0.715 | 0.786 | 0.798 | 0.702 | 0.708 | 0.787 | 0.797 |
| s = 4 | 0.710 | 0.718 | **0.787** | **0.799** | 0.707 | 0.715 | **0.788** | **0.799** |
| $s = \sqrt{2}\log{(C-1)}$ | **0.713** | 0.723 | 0.783 | 0.795 | **0.713** | 0.723 | 0.781 | 0.793 |
| $s = \sqrt{64}$ | 0.710 | **0.725** | 0.776 | 0.789 | 0.710 | **0.725** | 0.774 | 0.787 |
| s = 10 | 0.702 | 0.721 | 0.768 | 0.781 | 0.700 | 0.719 | 0.765 | 0.779 |
| $s = \sqrt{128}$ | - | 0.718 | - | 0.778 | - | 0.715 | - | 0.774 |

Table 3: The performance of different scales for 64 and 128 bits on different benchmark datasets. For $s = \sqrt{2}\log{(C-1)}$, ImageNet100 has $C = 100, s = 6.1793$ and MS-COCO has $C = 80, s = 6.4985$.

In the main paper, the loss function for learning to hash is:

$$L = -\frac{1}{N} \sum_{n=1}^{N} \log \frac{\exp\left(\sqrt{K}\cos\left(\theta_{y_n}\right)\right)}{\exp\left(\sqrt{K}\cos\left(\theta_{y_n}\right)\right) + \sum_{i=1, i \neq y_n}^{C} \exp\left(\sqrt{K}\cos\left(\theta_{ni}\right)\right)} \tag{7}$$

By default, we scale the cosine similarity to have a norm of $\sqrt{K}$ which follows the norm of binary codes (we use the symbol $s$ to represent the scale, e.g., $s = \sqrt{K}$). Nonetheless, the scale in [23, 5] does play an important role in the optimization, which affect the performance if the scale is not suitable. Furthermore, AdaCos [28] analyzed the importance of scale and showed how it can further improve the performance with adaptive scaling. Hence, we trained with a various range of scales, with a multiples of 2 and we summarize the effect of scales in Table 3. We didn't repeat the experiments for 3 times, but we run with the same seed for all different scales under different margin types, i.e., cosine margin (**OrthoCos+BN**) and angular margin (**OrthoArc+BN**), the margin $m$ is fixed as 0.2.

**Effect of scales.** We observe that all models often requires lower scale than $\sqrt{K}$ in order to achieve the best performance. While we were tweaking with different scales, we found that adaptive scale, $s = \sqrt{2} \log (C - 1)$ often produced the best (or closer to the best) performance. Hence, we conclude that in practice, we can follow the work done in AdaCos [28], by setting $s = \sqrt{2} \log (C - 1)$ instead of $s = \sqrt{K}$.

### D.3 Orthogonal Targets Generation

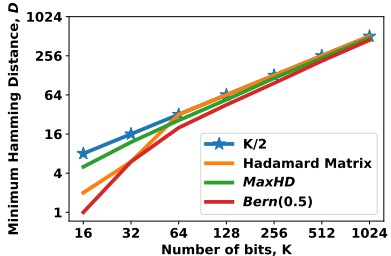

(a) Various number of bits with $C = 100$.

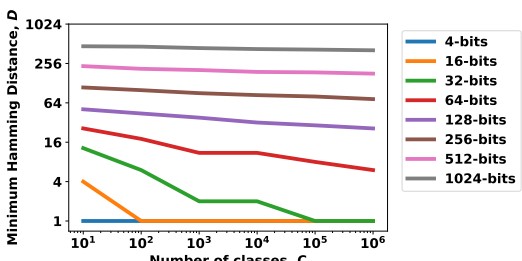

(b) Various number of classes and bits.

Figure 1: Minimum hamming distance between any two classes of binary orthogonal targets. The y-axis is in $\log_2$ scaling, since Hamming distance of 0 cannot be displayed properly in $\log_2$ scaling, therefore we set both Hamming distance of 0 and 1 as 1 for the display purpose.

| Orthogonal Targets | ImageNet100 (mAP@1K) | | | | NUS-WIDE (mAP@5K) | | | | MS COCO (mAP@5K) | | | |
|---|---|---|---|---|---|---|---|---|---|---|---|---|
| | 16 | 32 | 64 | 128 | 16 | 32 | 64 | 128 | 16 | 32 | 64 | 128 |
| Hadamard Matrix | 0.603 | **0.683** | **0.717** | **0.721** | 0.803 | 0.829 | 0.838 | 0.845 | **0.718** | **0.765** | **0.778** | 0.777 |
| *MaxHD* | **0.620** | 0.680 | 0.711 | 0.720 | **0.806** | **0.832** | 0.842 | **0.851** | 0.717 | 0.758 | **0.778** | 0.779 |
| Bernoulli Distribution | 0.608 | 0.679 | 0.711 | 0.718 | 0.804 | 0.830 | **0.845** | 0.850 | 0.706 | 0.759 | 0.776 | 0.777 |

Table 4: Performance of different methods for 4 different bits on different benchmark datasets. *MaxHD* refer to Maximum Hamming Distance method, see Algorithm 3. **Bold** values indicate best performance in the column.

We use different methods to generate targets, and then plot the *minimum* Hamming distance between orthogonal targets of any two classes against different number of bits ($K \in [4, 16, 32, 64, ..., 2048]$) in Figure 1a. With lower $K$ (i.e., $2K < C$), Hadamard Matrix relies on Bernoulli distribution [27], but with higher $K$ s.t. $2K \geq C$, the Hadamard Matrix guarantees the minimum Hamming distance is the maximal expectation of inter-class Hamming distance (i.e., $\frac{K}{2}$) because of the property of orthogonality in the Hadamard matrix. *MaxHD* (see Algorithm 3) generate targets with the objective of maximum inter-class Hamming Distance, even at lower bits, it has the largest minimum Hamming Distance (closer to $\frac{K}{2}$) and consistently larger than Bernoulli distribution $Bern(0.5)$ at any number of bits. Table 4 summarizes the performance with different methods of orthogonal targets generation. At lower bits, *MaxHD* performs the best due to the optimization of the objective (i.e., with 1% improvement in ImageNet100), but at higher bits, the improvement become negligible (e.g., less than 0.5% improvement).

In practice, we can simply use $Bern(0.5)$ to generate binary orthogonal targets. As shown in Figure 1b, at $K = 128$ is sufficient to handle a million classes $C = 1,000,000$, with minimum Hamming distance of $\sim 32$ between any two classes. To guarantee a better separability in such tremendous number of classes, we can increase $K$ to 256-bits or even higher, so that the minimum Hamming distance between any two classes will be higher.

### D.4 Multi-class Classification Losses

Our method uses only single classification objective, and the performance depends on the choice of classification loss. We noticed the uses of different loss methods have significant difference on multi-class classification, and thus we did an ablation study on how different losses affect the performance on multi-class datasets. In this study, we use 64-bits **OrthoCos+BN** with $m = 0.2$ and $s = \sqrt{K}$ on NUS-WIDE [3] and MS-COCO [12].

| Loss | Methods | NUS-WIDE (mAP@5K) | MS COCO (mAP@5K) |
|---|---|---|---|
| BCE | Sigmoid | 0.791 | 0.400 |
| | Sigmoid + Imbalance Weights | 0.827 | 0.717 |
| CE | Softmax + Label Smoothing [16] | **0.850** | **0.785** |

Table 5: Performance of different multi-class classification loss on two multi-class benchmark datasets with **64-bits OrthoCos+BN**. **BCE** denotes Binary Cross Entropy and **CE** denotes Cross Entropy. **Bold** values indicate best performance in the column.

We ran the experiments on three different methods (**Sigmoid**, **Sigmoid + Imbalance Weights**, and **Softmax + Label Smoothing [16]**) and Table 5 shows the results of these methods, we notice that our proposed method (**Softmax + Label Smoothing**) performs the best on both datasets with at most 38.5% improvements. We conclude that it is very important to apply imbalance weights or label smoothing in multi-labels datasets. Otherwise, a large number of negative classes will dominate the loss minimization. HashNet [2] also found this problem and solved with a imbalance mask to balance between positive and negative data pairs.

Note that JMLH [20] is using the first option by default (**Sigmoid**) which performs badly, hence we apply our method for JMLH in multi-labels datasets.

For imbalance weights, we simply only focus on the target and incorrect[17] classes, i.e., with scale of 1, while non-target classes are with lower scale, i.e., $1/C$. The imbalance weights are adaptive to different samples.

For label smoothing, see Algorithm 1 for details.

### D.5   Domain Shifting

| Mean and Variance | **GLDv2** (mAP@100) | | | $\mathcal{R}$**Oxf-Hard** (mAP@all) | | | $\mathcal{R}$**Paris-Hard** (mAP@all) | | |
|---|---|---|---|---|---|---|---|---|---|
| | 128 | 512 | 2048 | 128 | 512 | 2048 | 128 | 512 | 2048 |
| Unchanged | **0.035** | 0.107 | **0.149** | 0.015 | 0.036 | 0.135 | 0.048 | 0.159 | 0.406 |
| $\mu = 0, \sigma = 1$ | 0.025 | 0.100 | 0.145 | 0.158 | **0.376** | 0.437 | 0.397 | 0.606 | **0.675** |
| $\mu = 0, \sigma = \sigma_{new}$ | 0.025 | 0.100 | 0.145 | 0.158 | **0.376** | 0.437 | 0.397 | 0.606 | **0.675** |
| $\mu = \mu_{new}, \sigma = 1$ | 0.034 | **0.108** | **0.149** | **0.184** | 0.359 | **0.447** | **0.416** | **0.608** | 0.669 |
| $\mu = \mu_{new}, \sigma = \sigma_{new}$ | 0.034 | **0.108** | **0.149** | **0.184** | 0.359 | **0.447** | **0.416** | **0.608** | 0.669 |

Table 6: Performance of different mean and variance on Batch Normalize layer for 3 different numbers of bits on different instance-level benchmark datasets. **Bold** values indicate best performance in the column.

As the model is trained with GLDv2, the running mean and variance in the BN layer might experience domain shifting problem [10] when testing directly on different datasets (e.g., $\mathcal{R}$Oxf and $\mathcal{R}$Par). We empirically found that using running mean and variance from GLDv2 will lead to a large performance drop in Hamming distance retrieval. One simple solution is to recompute the mean and variance from all continuous codes in the database, then update the running mean and variance with the computed mean and variance. We first analyze the equation of BN during inference stage:

$$v = \frac{\gamma}{\sqrt{\sigma + \epsilon}} \cdot \hat{v} + (\beta - \frac{\gamma\mu}{\sqrt{\sigma + \epsilon}}) = \gamma\frac{\hat{v} - \mu}{\sqrt{\sigma + \epsilon}} + \beta, \tag{8}$$

in which $\hat{v}$ is the inputs (i.e., the continuous codes before normalization), $v$ is the outputs (i.e., the continuous codes after normalization), $\mu$ is the running mean, $\sigma$ is the running variance, $\gamma$ is the scale, $\beta$ is the bias and $\epsilon$ is an arbitrarily small positive quantity (e.g., $\epsilon = 10^{-7}$). Then we can compute the hash codes $b$ through a *sign* function, which outputs the value of $+1$ if $v > 0$, otherwise $-1$. We can see that the sign function ignore the magnitude of $v$. However, if we shift $v$ by $\mu$, i.e. $v_{\text{shift}} = v - \mu$, then we can see that if $\mu > v$, then $v_{shift}$ becomes negative and hence taking a negative code $-1$. Hence, the shifting will affect the performance of Hamming distance retrieval, we notice that using the learned scale and bias $\gamma, \beta$ and the running mean and variance $\mu, \gamma$ from GLDv2 will cause large performance drop in $\mathcal{R}$Oxf-Hard, and $\mathcal{R}$Par-Hard.

---

[17]predictions that are incorrect.

To fix the performance dropping, we first reset $\gamma$ and $\beta$ to 1 and 0 instead of trained values. Then, we set $\mu$ and $\sigma$ with the new computed mean and variance from $\hat{v}$ (i.e., the continuous codes before normalization), namely $\mu_{new}$ and $\sigma_{new}$. Finally, we evaluate with the new hash codes for mAP scores on GLDv2 [25], $\mathcal{R}$Oxf-Hard, and $\mathcal{R}$Par-Hard.

We have tried 4 different ways for the fix which are i) update both mean $\mu$ as 0 and variance $\sigma$ as 1; ii) update only variance $\sigma$ with $\sigma_{new}$ and mean $\mu$ remain 0; iii) update only mean $\mu$ with $\mu_{new}$ and variance $\sigma$ remain 1; iv) update both mean and variance with $\mu_{new}$ and $\sigma_{new}$.

The results are summarized in Table 6. We observe that there are huge improvement on the performances of both $\mathcal{R}$Oxf and $\mathcal{R}$Par, with at most 34.4% and 44.1%, by using the computed mean and variance from the respective datasets. There is no performance boost on GLDv2 dataset as it is in the same domain. We also observe that whether or not to update the variance $\sigma$ will not affect the performance because it will only affect the magnitude, which is ignored by the sign function.

### D.6 Visualization of Hash Codes

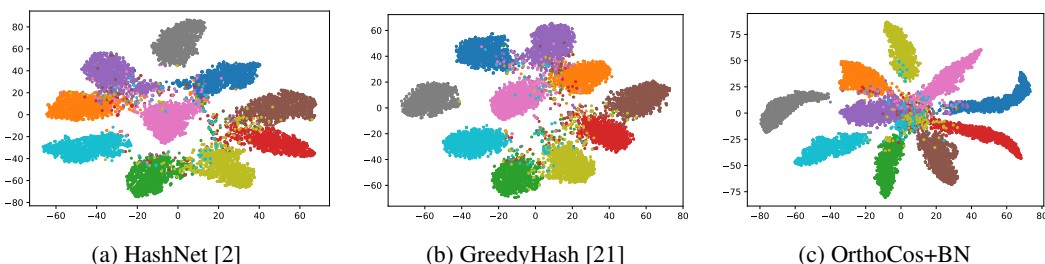

|     (a) HashNet [2]     |     (b) GreedyHash [21]     |     (c) OrthoCos+BN     |

Figure 2: t-SNE visualization with 10 random classes on 64-bits hash codes of ImageNet100.

In Figure 2, we plot the t-SNE visualization [22] on our method and two other methods to compare the quality of hash codes generated. We select 10 random classes from ImageNet100 and plot the t-SNE visualization using the hash codes from these classes. We can observe that the hash codes generated by our method is more compact, well separated and have more discriminant structure compare to HashNet[2] and GreedyHash[21]. Hence, the better quality of the hash codes result in a better and accurate image retrieval.

### D.7 The Separability of Hamming distances

We have selected few previous works and visualize the histogram of intra-class and inter-class Hamming distances with 64-bits ImageNet100.

To compare between the effectiveness of pair-wise, triplet-wise and point-wise, we have selected **HashNet** [2], **DTSH** [24] and **GreedyHash** [21] and plot them in Figure 3a, 3b and 3c respectively. It can be seen that the separability is improving from pair-wise (13.90) to triplet-wise (16.47) and finally point-wise (17.34), indicating the effectiveness of point-wise method in learning to hash.

Further, we compare between different code balance schemes, i.e., no code balance (**CE**), code balancing with BN (**CE+BN**) and code balancing with Bi-Half (**CE+BiHalf**) and they were plotted in Figure 3d, 3e and 3f respectively. Note that **CE** model perform the worst among all methods, the separability is lowest and the overlapping between the histogram of intra-class and inter-class is quite obvious. Although **CE+BN** shows a little improve in the separability, it can be seen that the histogram of inter-class distances become denser, indicating lesser overlapping and hence improve the performance. Lastly, **CE+BiHalf** has a proxy derivative to solve vanishing gradient problem, showing highest separability, hence perform the best among different code balance schemes.

To compare between pre-defined targets based methods (i.e., **DPN** [6] and **CSQ** [27]) with our method (**OrthoCos+BN**), they were plotted in Figure 3g, 3h and 3i respectively. It is clearly to see that there are a lot of Hamming distances of 0 within the same class, indicating lower intra-class variance in all 3 methods. We conclude that learning with pre-defined targets can result in more accurate hash codes.

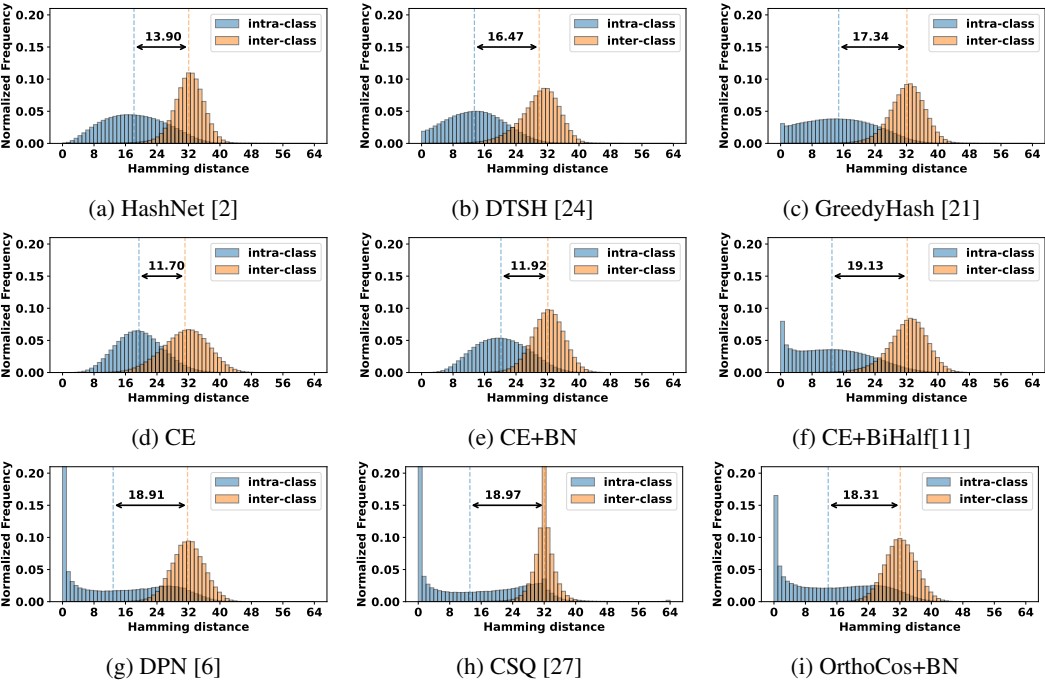

Figure 3: Histogram of intra-class and inter-class Hamming distances with 64-bits ImageNet100. The arrow annotation is the separability in Hamming distances, $\mathbb{E}[D_{inter}] - \mathbb{E}[D_{intra}]$. We normalized the frequency so that sum of all bins equal to 1.

# E Algorithm

Algorithm 1 summarizes our method, OrthoHash for learning to hash. Algorithm 2 summarizes how we compute for hash centers and orthogonality $\|\mathbf{HH^T} - \mathbb{I}\|$. Algorithm 3 summarizes how we generate the binary orthogonal targets with the objective of maximum inter-class Hamming distance heuristically.

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

**Algorithm 1:** PyTorch-style pseudocode for OrthoHash

```
# net:  backbone network
# latent:  hash latent layer
#
# o:  binary orthogonal target (CxK)
#  prob = torch.ones(C, K) * 0.5
#  o = torch.bernoulli(prob) * 2.0 - 1.0
#
# N: batch size
# C: number of classes
# q:  dimensionality of feature representations
# K: number of bits
# scale:  default is √K, but can be adjusted
#
# to_vec:  convert scalar-label(s) to one-hot vector
# mm:  matrix-matrix multiplication

for x, y in dataloader:
    # compute representations
    f = net(x) # Nxq

    # compute continuous codes
    v = latent(f) # NxK

    # convert to label vector
    # e.g.  [[1, 3], [0]] -> [[0, 1, 0, 1], [1, 0, 0, 0]]
    y_vec = to_vec(y) # NxC

    # l2 normalization on continuous codes and orthogonal target
    v_norm = v / v.norm(p=2, dim=1) # NxK
    o_norm = o / o.norm(p=2, dim=1) # CxK

    # compute cosine similarity
    cs_logits = mm(v_norm, o_norm.t()) # NxC
    # add cosine margin and scaling
    margin_logits = scale * (cs_logits - y_vec * margin)

    # label smoothing for multi-class
    if is_multiclass:
        y_vec = y_vec / y_vec.sum(dim=1) # NxC
        # e.g.  y_vec = [[0, 1, 0, 1], [1, 0, 0, 0]]
        # e.g.  y_vec.sum(dim=1) = [2, 1]
        # e.g.  new y_vec = [[0, 0.5, 0, 0.5], [1.0, 0, 0, 0]]

    # softmax cross entropy loss
    log_logits = log_softmax(margin_logits)
    loss = - (y_vec * log_logits).sum(dim=1).mean()

    # optimization step
    loss.backward()
    optimizer.step()
```

**Algorithm 2:** PyTorch-style pseudocode for Hash Centers Computation and Orthogonality

```
# N: number of data in database
# K: number of bits
# C: number of classes
#
# V: database continuous codes (NxK)
# Y: database labels (Nx1)
#
# mm:  matrix-matrix multiplication
# eye:  identity matrix

# compute binary hash codes
B = V.sign() # NxK
# initialize hash centers
H = zeros(C, K) # CxK
for i in range(C):
   # compute average hash codes for i-th class
   avg_B = B[Y == i].mean(dim=0) # K
   H[i] = avg_B.sign()
# Compute Orthogonality
ortho = (mm(H, H.t()) - eye(C)).norm(p=2)
```

**Algorithm 3:** PyTorch-style pseudocode for generating *MaxHD* orthogonal targets

```
# C: number of class
# K: number of bits
#
# maxtries:  10000
# initdist:  0.61
# mindist:  0.2
# reducedist:  0.01
#
# get_hd:  compute hamming distance between two vectors, and normalize
 output to 0-1

o = torch.zeros(C, K)
i = 0
count = 0
currdist = initdist

while i < C:
   # generate target through bernoulli distribution
   prob = torch.ones(K) * 0.5
   c = torch.bernoulli(prob) * 2.0 - 1.0
   nobreak = True

   # to compare distance with previous classes
   for j in range(i):
      # if target satisfies constraint
      if get_hd(c, o[j]) < currdist:
         i -= 1
         nobreak = False
         break

   # if successfully found a target
   if nobreak:
      o[i] = c
   else:
      count += 1

   # if not able to search a target
   if count >= maxtries:
      count = 0
      # decrease the constraint
      currdist -= reducedist
      # reach constraint limit
      if currdist < mindist:
         raise ValueError('Cannot find target')
   i += 1

# shuffle the orthogonal targets
o = o[torch.randperm(C)]
```