# OpenReview forum: "One Loss for All: Deep Hashing with a Single Cosine Similarity based Learning Objective"
_NeurIPS.cc/2021/Conference — NeurIPS 2021 Poster_

### Official Review · Reviewer_MNiD · 2021-07-14

**Rating:** 7
**Confidence:** 5

**Summary:**

This paper presents OrthoHash, a new deep hashing model that is trained by maximizing the cosine similarity between the continuous codes and the corresponding binary orthogonal target. In addition, code balancing is achieved by using batch normalization. Different from prior works that typically involves multiple loss functions, the training objective in this paper is simpler and easy to optimize.

**Limitations And Societal Impact:**

Yes

**Main Review:**

Strength:
- Neat idea and good novelty: the proposed training objective is well designed with the idea of cosine similarity. The final formulation with softmax-like loss function is very interesting, and it is simpler than the existing multi-loss approaches. Additionally, the use of BN for balancing code is elegant.
- Strong results on multiple datasets. It is highly competitive compared to multi-loss approaches.
- Good ablation studies to demonstrate the performance impact of different factors.
- The paper is well-written and easy to understand.

Weakness:
- One possible weakness is that it may have large performance degrade when the bit length is short. That is because in the target code generation, the target is sampled from Bern(0.5). One may expect larger information loss with short code.
- Evaluation: For ROxf and RPar, the authors use multi-scale features for instance retrieval. It is unclear to me if the compared exiting deep hashing methods also use such a multi-scale scheme. Does it a fair comparison?
- Evaluation: The authors use mAP@1K for ImageNet100, but mAP@5k for NUS-WISE and COCO. In instance retrieval, the authors use mAP@100 and mAP@all instead. They are inconsistent and not easy to interpret the performance. The authors may want to provide some intuitions/discussions about their design choices. For example, how many returned images are required to better reflect the  retrieval performance?  What's the effect of evaluating all returned images?
- In current presentation, the proposed method performs only on par with prior works in category-level retrieval. This gives an impression that only ordinary improvements are achieved. I believe this paper has many advantages that are not well demonstrated in current form. One shining point should be its scalability to very large dataset. I was hopping to see at least one experiment at a large scale, to clearly differentiate itself from pair-wise and triplet-wise methods. This would strengthen the paper significantly.


Overall:
- Overall this is a nice paper. My main concerns are in their evaluation. But this paper has merits that outweigh the flaws.
- This paper presents a simpler learning objective that can reduce the difficulty of training deep hash models. Given its effectiveness and the simplicity, I think this paper could serve as a strong baseline to facilitate future research. In my opinion, it would bring insightful contributions to our community. My initial recommendation is Accept.

**Time Spent Reviewing:**

12

---

> ### Author Response · Authors · 2021-08-07
> **Reply to Reviewer MNiD**
>
> $\textbf{Larger loss with short code.}$ A very short code indeed challenges most hashing methods. However, in the Supplementary Material, Section D.3 (Page 6), we have shown in Table 4 that explicitly maximizing inter-class hamming distance (maxHD) or Hadamard matrix style for target code generation performs better than Bernoulli with short code (e.g. 16 bits in ImageNet100). We plan to examine more carefully the impact of  (extremely-)short code (e.g. $<8$ bits) as part of our future work.
>
> $\textbf{Evaluation Protocol.}$ All of our evaluation protocols are strictly followed all previous reported works [1,2,3,4,5,6,7,8] in order to have a fair comparison as detailed in our Supplementary Material, Section C.3 - Evaluation Detail. We will make it clearer.
>
> More specifically, following the settings of prior works on category-level retrieval,  we used mAP@1000 for ImageNet100, mAP@5K for NUSWIDE, and MSCOCO. To make it more consistent,  we now have conducted new evaluations on mAP@[10,100,1000,10000,all] of 64-bits ImageNet100, NUSWIDE \& MSCOCO of DTSH (perform well for multi-labeled), CSQ (perform well for single-labeled) and our method (OrthoCos+BN) for category-level retrieval. The results are recorded below. It is observed that our method still outperforms SOTAs under these mAP metrics. Although our method is only on par with CSQ and DTSH in mAP@all, it is worth noting that in mAP@all, the order of all retrieved images are being considered altogether, in contrast for mAP@1000, only the order of first 1000 retrieved are being considered. Hence, the protocol (i.e. mAP@1K for ImageNet100, mAP@5k for NUS-WISE and MSCOCO) set by SOTAs [1,2,3,4,5,6,7,8] are reasonable in this case.
>
> | 64-bits ImageNet100 | mAP@10 | mAP@100 | mAP@1000 | mAP@10000 | mAP@all |
> | ------------------- |:------:| -------:| -------- | --------- | ------- |
> | DTSH                | 0.667  | 0.631   | 0.585    | 0.503     | 0.480   |
> | CSQ                 | 0.649  | 0.706   | 0.696    | 0.655     | 0.608   |
> | OrthoCos+BN         | 0.720  | 0.721   | 0.711    | 0.664     | 0.608   |
>
> | 64-bits NUSWIDE | mAP@10 | mAP@100 | mAP@1000 | mAP@10000 | mAP@all |
> | ------------------- |:------:| -------:| -------- | --------- | ------- |
> | DTSH                | 0.884 | 0.865   | 0.856    | 0.841     | 0.740   |
> | CSQ                 | 0.884  | 0.864   | 0.852    | 0.827     | 0.714   |
> | OrthoCos+BN         | 0.892  | 0.874   | 0.864   | 0.839     | 0.712   |
>
> | 64-bits MSCOCO | mAP@10 | mAP@100 | mAP@1000 | mAP@10000 | mAP@all |
> | -------------- |:------:| -------:| -------- | --------- | ------- |
> | DTSH           | 0.915  | 0.824   | 0.786    | 0.736     | 0.666   |
> | CSQ            | 0.934  | 0.859   | 0.821    | 0.760     | 0.644   |
> | OrthoCos+BN    | 0.943  | 0.871   | 0.830    | 0.763     | 0.636   |
>
> For the evaluation metrics employed in the instance-level retrieval, we also strictly followed all previous reported works [9,10]. This task is more difficult, e.g., the number of images per instance have lesser than 100 images in GLDv2. For GLDv2, we retrieve 100 images from the database which contain 762K images, and note that there will be about 100K number of unique landmarks, hence measuring mAP@100 seem reasonable. Since ROxf/RPar contain only 5000 to 6000 images (with about 11 unique landmarks), mAP@all seem reasonable too. Note that, we did not include ROxf/RPar into training;  we only used GLDv2-trained model run query for ROxF/RPar.
>
> $\textbf{Only ordinary improvements are achieved.}$ Please refer to the answer from "(4) Results explanation in Table 1" in the reply to Reviewer 7uiu.
>
> $\textbf{Scalability to very large dataset.}$ Thanks. At the moment,  all of our experiments in category-level is around 130K images and for instance-level retrieval (i.e., GLDv2), the database images are about 762K. These are reasonable scale and used by most of the reported work too. We agree that our method may shine even more with larger scale benchmarks. However,   due to the limited computation resources at academia and limited time, we can only leave it to future work.
>
> $\textbf{Reference:}$
> 1. Zhangjie Cao, Mingsheng Long, Jianmin Wang, and Philip S. Yu. Hashnet: Deep learning to hash by continuation. ICCV 2017.
> 2. Lixin Fan, Kam Woh Ng, Ce Ju, Tianyu Zhang, and Chee Seng Chan. Deep polarized network for supervised learning of accurate binary hashing codes. IJCAI 2020.
> 3. Shupeng Su, Chao Zhang, Kai Han, and Yonghong Tian. Greedy hash: Towards fast optimization for accurate hash coding in cnn. NeurIPS 2018.
> 4. Haomiao Liu, Ruiping Wang, Shiguang Shan, and Xilin Chen. Deep supervised hashing for fast image retrieval. CVPR 2016.
> 5. Yuming Shen, Jie Qin, Jiaxin Chen, Li Liu, Fan Zhu, and Ziyi Shen. Embarrassingly simple binary representation learning. ICCV Workshop 2019.
> 6. Rongkai Xia, Yan Pan, Hanjiang Lai, Cong Liu, and Shuicheng Yan. Supervised hashing for image retrieval via image representation learning. AAAI 2014.
> 7. Hanjiang Lai, Yan Pan, Ye Liu, and Shuicheng Yan. Simultaneous feature learning and hash coding with deep neural networks. CVPR 2015.
> 8. Li Yuan, Tao Wang, Xiaopeng Zhang, Francis EH Tay, Zequn Jie, Wei Liu, and Jiashi Feng. Central similarity quantization for efficient image and video retrieval. CVPR 2020.
> 9. Bingyi Cao, André Araujo, and Jack Sim. Unifying deep local and global features for image search. ECCV 2020.
> 10. Filip Radenovic, Ahmet Iscen, Giorgos Tolias, Yannis Avrithis, and Ondrej Chum. Revisiting oxford and paris: Large-scale image retrieval benchmarking. CVPR 2018.

---

### Official Review · Reviewer_7uiu · 2021-07-17

**Rating:** 7
**Confidence:** 3

**Summary:**

In this paper, authors propose a novel deep hashing model with only a single learning objective and claim that state of the art papers generally use a large number (>4) of losses. Specifically, authors show that maximizing the cosine similarity between the continuous codes and their corresponding binary orthogonal codes can ensure both hashcode discriminativeness and quantization error minimization. Authors show significance of their method using extensive experiments.


**Limitations And Societal Impact:**

I don't see any potential negative societal impact of their work.

**Main Review:**

1) Main claims made in the abstract and introduction accurately reflect the paper's contributions and scope. Proposed method is simple but novel. The submission is technically sound and well supported by experimental results.
2) Under what conditions the proposed method would fail? Can authors comment more on this?
3) Using orthogonal targets is a simple but clever idea. Can authors comment on how can modify this method when one doesn’t have categorical labels but have some pairwise constraints and the goal is to learn quantized representation learning for better retrieval?
4) In Table 1, proposed method is not always the best. From the discussion it was not clear why? Can authors comment more on why? Is it because the state of the art methods were already close to the best possible results or there is some drawback of the method that does not improve the results on some datasets?
5) Angular margin used in the paper seems very related to semantic hashing [1] where angular similarity was used to generate representations better for hashing retrieval. Can authors compare and comment?
6) In Figure 6, not only the distance between two distributions (intra class and inter class) matter but also the overlap between them matters. Can authors calculate and comment on the overlap between two distributions?

[1] https://openreview.net/pdf?id=sFDJNhwz7S

**Time Spent Reviewing:**

4

---

> ### Author Response · Authors · 2021-08-07
> **Reply to Reviewer 7uiu**
>
> $\textbf{(2) Under what conditions the proposed method would fail?}$ One such condition is when the number of bits is too small (<8 bits), especially when number of classes is much greater than the number of bits. In this case, there will be overlapping in the generated target code (i.e., the number of maximum unique codes is equal to $2^K$ where $K$ is number of bits). Overcoming this limitation is part of the  future work. We will add this discussion.
>
> $\textbf{(3) Modification on our proposed method with pairwise constraints.}$ Good questions! We think it is possible to treat this problem as instance-level retrieval (e.g., image matching [2]) where every pair are considered as a single instance pair. This way, we can treat each instance (e.g., if pair A <-> B, and pair B <-> C. As such, we can possibly deduce that all A, B, C are the same instance) as a single class and perform the experiments like how we did in Section 4.2 (instance-level retrieval).
>
> $\textbf{(4) Results explanation in Table 1.}$ Thank you. We think it is the combination of both factors:
> a) The category-level retrieval task is relatively easy and the latest results reported on both datasets are clearly saturating. This explains the small gaps between different methods in Table 1.  We therefore tested our method on the more difficult  instance-level retrieval task (Section 4.2 and Table 2). For this task the advantage of our method over the SOTA alternatives is much clearer.
> (b) NUS-WIDE/MS-COCO are multi-labeled data (and with 21/80 classes), hence it might be easier for pairwise/triplet-wise losses based methods. For them, as long as one of the labels appeared in both query and retrieved images, it is considered as positive, otherwise negative. The retrieval evaluation follows exactly the same setting, therefore favors these pairwise/triplet-wise methods. In contrast, our method treats multi-labeled as a classification problem with  harder constraints on  the output having an equivalent probability for the labeled classes based on the label-smoothing concept. This mismatch in the training objective and test setting thus gives our method a disadvantage.
>
> $\textbf{(5) Relation with semantic hashing [1].}$ Great suggestion! We will cite and compare with this work as suggested. These two works are indeed related but also have vital differences. More specifically, the method in [1] is based on the probability of collision under SimHash [4],  and describes the hashing problem under pairwise constraint. In contrast, our method is pointwise. In the loss design our single loss is also very different from theirs.
> Overall, we think pointwise method would be better than pairwise/triplet-wise based methods in learning to hash.
> This is because  pairwise/triplet-wise based methods need to solve a sample mining problem as well (also discussed in [1]), making the optimization problem harder.
>
> $\textbf{(6) Overlap between two distributions in Figure 6.}$ Thank you. We believe you meant Figure 3, as we do not have a Figure 6 in our paper and supplementary. As requested, we measured the overlapping by using histogram intersection, and the results are 0.2365, 0.2049 and 0.2586 for Figure 3a,b,c respectively. However, this does not represent the true performance as these overlapping area were happened mostly at the region of inter-class (specifically, at hamming distance of 16 to 40), where this distance are used to positioned the retrieved images in end of the retrieval.
>
> Therefore, we consider it more appropriate to look into on those regions where the hamming distance ranges between 0 to 15. In this way, other than overlapping area, we also measure the sum of area as well. The sum of area (the bins of intra-class only) are 0.3946, 0.5414 and 0.5613 and the overlapping are 0.0024, 0.0029 and 0.0014 for Figure 3a,b,c respectively. As we can see, while the overlapping area is too insignificant to verify, the sum of area from 0 to 15 are proportional to both the distance between two distributions (i.e., 13.90, 17.34 and 18.31) and the performance (i.e., 0.573, 0.659 and 0.711). While the exact performance may be explained by multiple factors, we believe that the distance between two distributions in Figure 3 can help explain the superiority of our method.
>
> $\textbf{Reference:}$
> 1. Levi Boyles, Aniket Anand Deshmukh, Urun Dogan, Rajesh Koduru, Charles Denis, Eren Manavoglu. Semantic Hashing with Locality Sensitive Embeddings. https://openreview.net/forum?id=sFDJNhwz7S
> 2. https://image-matching-workshop.github.io/
> 3. Ye, Mang, Jianbing Shen, Gaojie Lin, Tao Xiang, Ling Shao, and Steven CH Hoi. Deep learning for person re-identification: A survey and outlook. TPAMI 2021.
> 4. Moses S Charikar. Similarity estimation techniques from rounding algorithms. In Proceedings of the thirty-fourth annual ACM symposium on Theory of Computing, pages 380–388, 2002.

---

### Official Review · Reviewer_N6CJ · 2021-07-20

**Rating:** 4
**Confidence:** 4

**Summary:**

The paper proposes a deep hashing model with only a single learning objective. It avoids the problem that existing deep hashing models are difficult to train due to a large number (>4) of losses. Specifically, it maximizes the cosine similarity between the continuous codes and their corresponding binary orthogonal codes to ensure both the discriminative capability of hash codes and the quantization error minimization. Besides, it adopts a Batch Normalization layer to ensure code balance and leverages the Label Smoothing strategy to modify the Cross-Entropy loss to tackle multi-labels classification. Extensive experiments show that the proposed method achieves better performance compared with the state-of-the-art multi-loss hashing methods on several benchmark datasets.

**Limitations And Societal Impact:**

1. According to the experimental results in Table 1, the proposed method does not consistently outperform the baseline methods, which is inconsistent with the sentence"outperforming the state-of-the-art multi-loss hashing models on three large-scale instance retrieval benchmarks, often by significant margins" mentioned in Abstract. In addition, the performance improvement of the proposed method is not significant.

2. As the main contribution of this work is to propose a unified objective function, to verify its superiority, the authors should compare the results of the following two experiments:
a) Baseline methods without any change
b) The variant baselines which replace the multiple loss functions with the proposed single loss

3. There are several issues that the authors may want to explain.
a) In the second paragraph of section 3 OrthoHash: One Loss for All, the authors write "o_i denotes a column vector belongs to i-th class". Since the dimension of the binary orthogonal targets O is C×K, I think o_i is a row vector.
b) The proposed method is supervised. Please expain the reason why using the unsupervised Bihalf as the baseline?
c) Why mAP@1k is used for ImageNet100 and mAP@5k for MS COCO for performance evaluation, especially when the two datasets are close in size. Besides, the metrics of performance evaluation for instance-level benchmark datasets are also different.

4. For experimental evaluation with SOTA methods, the authors should compare more newly deep hashing methods in the experiments.

**Main Review:**

The paper proposes a deep hashing model with only a single learning objective. It avoids the problem that existing deep hashing models are difficult to train due to a large number (>4) of losses. Specifically, it maximizes the cosine similarity between the continuous codes and their corresponding binary orthogonal codes to ensure both the discriminative capability of hash codes and the quantization error minimization. Besides, it adopts a Batch Normalization layer to ensure code balance and leverages the Label Smoothing strategy to modify the Cross-Entropy loss to tackle multi-labels classification. Extensive experiments show that the proposed method achieves better performance compared with the state-of-the-art multi-loss hashing methods on several benchmark datasets.

Strengths:
1. This work is meaningful for effectively optimizing the deep hashing models. It unifies the training objectives of deep hashing under a single classification objective.
2. The motivation is well presented by the authors in the manuscript.
3. The paper carries out extensive experiments and the supplementary material is rich in content.

Weaknesses:
1. According to the experimental results in Table 1, the proposed method does not consistently outperform the baseline methods, which is inconsistent with the sentence"outperforming the state-of-the-art multi-loss hashing models on three large-scale instance retrieval benchmarks, often by significant margins" mentioned in Abstract. In addition, the performance improvement of the proposed method is not significant.

2. As the main contribution of this work is to propose a unified objective function, to verify its superiority, the authors should compare the results of the following two experiments:
a) Baseline methods without any change
b) The variant baselines which replace the multiple loss functions with the proposed single loss

3. There are several issues that the authors may want to explain.
a) In the second paragraph of section 3 OrthoHash: One Loss for All, the authors write "o_i denotes a column vector belongs to i-th class". Since the dimension of the binary orthogonal targets O is C×K, I think o_i is a row vector.
b) The proposed method is supervised. Please expain the reason why using the unsupervised Bihalf as the baseline?
c) Why mAP@1k is used for ImageNet100 and mAP@5k for MS COCO for performance evaluation, especially when the two datasets are close in size. Besides, the metrics of performance evaluation for instance-level benchmark datasets are also different.

4. For experimental evaluation with SOTA methods, the authors should compare more newly deep hashing methods in the experiments.

**Time Spent Reviewing:**

6

---

> ### Author Response · Authors · 2021-08-07
> **Reply to Reviewer N6CJ**
>
> We would first like to thank the reviewer for the constructive suggestions.
>
> $\textbf{(1) Inconsistent with sentence and Table 1 in abstract.}$ We would like to point out that Table 1 compares category-level retrieval results, while the statement in abstract is on our method's superior performance on instance-level retrieval benchmarks. The results in Table 2 do show that our single-loss method clearly beat SOTA multi-loss alternatives on instance-level retrieval. We will further clarify.
>
> $\textbf{(2a) Baseline without change.}$ Thank you but that's exactly what we did. Specifically, all our baseline experiments in Table 1 and Table 2 were conducted with the baseline methods without any change, with the only exceptions being SDH-C and CE (Table 1). SDH [5] was first proposed in 2015 and they were using their own designed architecture (only 3 layers as backbone, yet using GIST descriptor as input), however, we think that the multiple loss functions (e.g., quantization, orthogonality and bit balancing) proposed in SDH are worth to compare, hence we modified the method such that it is using AlexNet as backbone and using classification objective instead of pairwise objective.
> As for CE, we also included CE+BN and CE+BiHalf to make it more competitive.
> Finally, for a fair comparison (as opposed to previous works which often have inconsistent hyperparameters setups), instead of taking the original results from the original paper, we ran all methods with AlexNet as backbone, Adam as optimizer and learning rate of 0.0001. We even used the same categories of ImageNet100 for all methods, as most papers did not mentioned what random categories they have chosen for evaluation. This explains the performance discrepancies to the reported results in the original papers.
>
> $\textbf{(2b) Variant baseline which replace multi-loss functions with proposed single loss.}$ Thanks. That's actually how we implemented our method with the proposed single loss -- we take an existing model and replace its multiple losses with our single one. For instance, our OrthoCos is based on CSQ [1]. The original CSQ model has no classification objective, but contains hash center as code target. We thus replace the CSQ loss functions with our single loss function resulting in our OrthoCos. We will make that clearer.
>
> $\textbf{(3a) Typo in the section 3.}$ Thanks for pointing it out.  $o_i$ is indeed a column vector. A corrected version of the sentence is $o_i \in [o_1, \dots ,o_C]^\intercal = O\in\\{-1,+1\\}^{C \times K}$. We will correct all these issues.
>
> $\textbf{(3b) Why unsupervised Bihalf layer is used in supervised learning.}$ Indeed,  Bihalf was initially proposed by [3] to solve unsupervised hashing problem. In [3], the Bihalf layer was appended into a FC layer to balance the output of every bit and solve the unsupervised hashing with unsupervised loss from [4]. Such a balance is also desirable under the supervised setting. Therefore,  in our experiment, we adopt the Bihalf layer in our supervised learning model. We will make it clearer.
>
> $\textbf{(3c) Evaluation Metric/Protocol.}$ Please refer to the answer from "Evaluation Protocol." in the reply to Reviewer MNiD.
>
> $\textbf{(4) Should compare more newly deep hashing methods in the experiments.}$ It is worth noting that the latest and popular baseline methods were indeed selected in Tables 1 and 2 at the time of submission. For instance, we have included the latest work from CVPR2020 (CSQ [1]), IJCAI2020 (DPN [2]) and AAAI2021 (Bihalf [3]). We will try to include newer works if accepted.
>
> $\textbf{Reference:}$
> 1. Li Yuan, Tao Wang, Xiaopeng Zhang, Francis EH Tay, Zequn Jie, Wei Liu, and Jiashi Feng. Central similarity quantization for efficient image and video retrieval. CVPR 2020.
> 2.  Lixin Fan, Kam Woh Ng, Ce Ju, Tianyu Zhang, and Chee Seng Chan. Deep polarized network for supervised learning of accurate binary hashing codes. IJCAI 2020.
> 3. Yunqiang Li and Jan van Gemert. Deep unsupervised image hashing by maximizing bit entropy. AAAI 2021.
> 4. Shupeng Su, Chao Zhang, Kai Han, and Yonghong Tian. Greedy hash: Towards fast optimization for accurate hash coding in cnn. NeurIPS 2018.
> 5. Venice Erin Liong, Jiwen Lu, Gang Wang, Pierre Moulin, and Jie Zhou. Deep hashing for compact binary codes learning. CVPR 2015.

---

### Decision · Program_Chairs · 2021-09-27

**Decision:**

Accept (Poster)

**Comment:**

Authors propose a novel deep hashing model with only a single learning objective which is a simplification from most state of the art papers generally use lots of losses and regularizer. Two of the three reviewers liked its effectiveness and the simplicity, they believe it would bring insightful contributions to our community. Reviewers liked the final formulation with softmax-like loss function. They also found the use of Batch normalization for balancing code elegant.

One of the reviewers raised certain concerns.  The authors in rebuttal clarified that most of the negative argument raised by reviewers are misunderstanding and are not valid. It is clear that the concerns raised were already addressed in the main draft.
.
Overall, the merits of the paper justifies the publication.